# COMPLEXITY OF FORMAL EXPLAINABILITY FOR SEQUENTIAL MODELS

## ABSTRACT

This work contributes to formal explainability in AI (FXAI) for sequential models, including Recurrent Neural Networks (RNN), Transformers, and automata models from formal language theory (e.g. finite-state automata). We study two common notions of explainability in FXAI: (1) abductive explanations (a.k.a. minimum sufficient reasons), and (2) counterfactual (a.k.a. contrastive) explanations. To account for various forms of sequential data (e.g. texts, time series, and videos), our models take a sequence of rational numbers as input. We first observe that simple RNN and Transformers suffer from NP-hard complexity (or sometimes undecidability) for both types of explanations. The works on extraction of automata from RNN hinge on the assumption that automata are more interpretable than RNN. Interestingly, it turns out that generating abductive explanations for DFA is computationally intractable (PSPACE-complete), for features that are represented by regular languages. On the positive side, we provide a polynomial-time algorithm that compute counterfactual explanations for deterministic finite automata (DFA). However, DFA are a highly inexpressive model for classifying sequences of numbers. To address this limitation, we extend our polynomial-time algorithm to two existing extensions of finite automata: (1) deterministic interval automata, and (2) deterministic register automata with a fixed number of registers. Automata learning algorithms already exist for both (1) and (2).

## 1 INTRODUCTION

Explainability (cf. Gunning et al. (2019); Molnar (2022); Linardatos et al. (2021)) has become an indispensable topic in machine learning. Various stakeholders (most notably, the EU regulators) are pushing for explainability for AI systems, particularly those that are used in critical decision making process (e.g. credit scoring and border control systems). Recent years have witnessed rise of *Formal Explainability in AI (FXAI)*, e.g., see Arenas et al. (2022); Barceló et al. (2020); Cooper & Marques-Silva (2023); Huang et al. (2023); Izza et al. (2022); Ignatiev et al. (2022); Shrotri et al. (2022); Marques-Silva (2021); Arenas et al. (2021); Marques-Silva (2022); Marques-Silva & Ignatiev (2023). Indeed, the ability to produce an explanation of the outputs of an AI system with a mathematically provable guarantee is becoming increasingly critical in many scenarios, e.g., in a legal setting and for safety-critical systems. Distinguishing features of FXAI are, among others, the use of logic and computational complexity (e.g. see Marques-Silva (2022); Marques-Silva & Ignatiev (2023) for an excellent survey).

Hitherto, the primary focus of FXAI has been on *non-sequential models*, i.e., where the length of the inputs is fixed. Such models include multilayer perceptrons, single-layer perceptrons, decision trees (or more generally, free binary decision diagrams), and decision lists/sets. Here (e.g. see Barceló et al. (2020); Izza et al. (2020; 2022); Huang et al. (2021); Ignatiev & Marques-Silva (2021); Marques-Silva & Ignatiev (2023)) models like decision trees and single-layer perceptrons have been shown to admit PTIME computational complexity for various explainability queries, in contrast to models like multilayer perceptrons, where the same queries were shown to be intractable (at least NP-hard). However, existing works in FXAI do not yet cover *sequential models*, i.e., where the length of the inputs can be arbitrary (e.g. texts, time series, videos). Such models include Recurrent Neural Networks (RNN) Rumelhart et al. (1985), Transformers Vaswani et al. (2017), and models from formal language theory like finite automata Weiss et al. (2018); Okudono et al. (2020); Wang et al.

(2017); Jacobsson (2005); Bollig et al. (2022). This paper initiates the study of FXAI for sequential models.

Two types of explanations have been singled out in FXAI Marques-Silva (2022); Marques-Silva & Ignatiev (2023): (1) *abductive* explanations, and (2) *counterfactual* (a.k.a. *contrastive*) explanations. The latter has a different name in the literature, e.g., *Minimum Change Required* Barceló et al. (2020). Both types of explanations fall within the category of local post-hoc explanations (e.g. see Molnar (2022)), i.e., an explanation for a specific input $v$ to a trained model $M$. Intuitively, given features $F_1, \ldots, F_n$ which are satisfied by $v$, an abductive explanation asks for a minimal subset $S \subseteq \{F_1, \ldots, F_n\}$ of these features such that $M$ produce the same output as $v$ on all inputs that satisfy $S$. As an example, suppose that $M$ was trained to classify mammals, $v$ signifies "humans", and $F_1$ indicates "lives on land". An abductive explanation could drop $F_1$ because there are mammals that does not satisfy $F_1$ (e.g. whales). On the other hand, a counterfactual explanation asks for an input $w$ that is "similar" to $v$ (i.e. requires a "minimal" amount of changes) so that the outputs of $M$ on $v$ and $w$ differ. For example, if $v$ indicates "bat", then $w$ could indicate "birds".

We study the problem of generating abductive and counterfactual explanations for sequential models, including Transformers, RNNs, and various models from automata theory. Since we deal with inputs of unbounded lengths, to define a "feature" it is crucial to have a finite representation of a set of sequences. One natural and common finite representation of a set of sequences from formal language theory is a regular language. Thus, we will use *regular languages to represent features*, since they are known to strike a good balance between expressivity and amenability to algorithmic analysis. For example, in a sentiment analysis of a book review at Amazon, a relevant feature could indicate whether the string "excellent" or "outstanding" occurs as a substring. Such features can be easily represented as regular languages. For counterfactual explanations, we furthermore incorporate a notion of "similarity" by comparing sequences with respect to three distance measures: Hamming distance, edit distance, and Dynamic Time Warping (DTW). These different measures are used in different application domains (e.g. information theory van Lint (1998), biological sequences Compeau (2014), and time series Nielsen (2019)).

We begin by observing that generating abductive and counterfactual explanations for popular sequential models (i.e. RNN and Transformers) is either intractable (at least NP-hard) or undecidable. The proof of these negative results can be achieved by straightforward (but sometimes slightly tedious) applications of Turing-completeness of RNN Siegelmann & Sontag (1995) and Transformers Pérez et al. (2021), or polynomial-time reductions from intractability of producing counterfactual explanations for multi-layer perceptrons Barceló et al. (2020).

Although abductive and counterfactual explanations are computationally difficult to generate for multi-layer perceptrons[1], the same problems were shown to be solvable in polynomial-time for decision trees and perceptrons Barceló et al. (2020); Izza et al. (2020); Huang et al. (2021); Izza et al. (2022). This prompts us to find sequential models that permit poly-time solutions for abductive/counterfactual explanations. For sequences over a finite alphabet, it has been hinted by the numerous available works on automata extraction from RNN (e.g. Dong et al. (2020); Weiss et al. (2018); Okudono et al. (2020); Wang et al. (2017); Jacobsson (2005); Bollig et al. (2022)) that Deterministic Finite Automata (DFA) might serve as such a model. In particular, in addition to the many desirable closure and algorithmic properties Sipser (1997) that DFA admit, there are many well-known learning algorithms for DFA over various learning settings, the prominent ones being Gold's DFA learner from examples Gold (1978) and Angluin's DFA learner $L^*$ through queries Angluin (1987a;b). Both Gold's and Angluin's algorithms have been subject to numerous further improvements in subsequent decades, whereby efficient and robust implementations exist (e.g. see the popular automata learning framework LearnLib Isberner et al. (2015); Steffen et al. (2011)). Among others, Angluin's $L^*$ algorithm has been used by Weiss et al. (2018) in the DFA extraction of RNN.

Interestingly, it is not hard to show that an abductive explanation for DFA with features represented by regular languages is computationally intractable (more precisely, PSPACE-complete). Our first main contribution is a polynomial-time algorithm for finding a counterfactual explanation for DFA with respect to Hamming distance, edit distance, and DTW. Despite this first positive result, when dealing with numeric data (i.e. sequences of numbers), DFA can admit only finitely many different

---

[1]This statement for counterfactual queries was proven in Barceló et al. (2020). For abductive explanations, the technique for proving hardness for *minimum sufficient reasons* in Barceló et al. (2020) can be used, but one reduces from computing a *prime implicant* of a boolean formula, which is NP-hard.

input numbers. This assumption is not valid for applications like time-series analysis, among others. For this reason, we study two well-known extensions of DFA from formal language theory: (1) deterministic interval automata D'Antoni & Veanes (2021), and (2) deterministic register automata Bojanczyk (2019). Extensions of classical learning algorithms for DFA to (1) and (2) have been active research directions in the past decade, e.g., see Moerman et al. (2017); Howar et al. (2012); Cassel et al. (2016); Drews & D'Antoni (2017); Argyros & D'Antoni (2018).

An interval automaton is specific type of "symbolic automaton" D'Antoni & Veanes (2021) that allows an interval as a label of a transition. For example, a transition $(p, [7, 10], q)$ will be interpreted as follows: go from state $p$ to state $q$, while consuming *any* element $e \in [7, 10]$ in the input tape. Expressively, this addresses the limitation of DFA that only finitely many numbers are allowed in the input, and can capture interesting concept (see Example 2 on stock trading signal). On the other hand, an interval automaton cannot compare numbers at different positions in the input. For this reason, we also consider register automata, which extend finite automata by finitely many registers, which can store numbers in the input sequence into a register, and compare them at a later point. More precisely, we consider deterministic $k$-register automata Bojanczyk (2019); Moerman et al. (2017) (i.e. with $k$ registers), where (in)equality comparisons are permitted. The model of $k$-register automata strictly extends interval automata, and is sufficiently powerful for modelling important concepts on time series (e.g. increasing/decreasing sequence of numbers, and the lowest weekly stock price constantly increases). See Example 3 for another stock trading signal. Our main result is that interval automata and deterministic $k$-register automata permit poly-time solutions for counterfactual explanations, whenever $k$ is fixed. Our polynomial-time algorithms for generating counterfactuals are especially interesting owing to the available learning algorithms for interval automata and deterministic $k$-register automata, which could pave way for interpretable models over time-series data. The proof of our positive result goes via a reduction to minimizing the values of "weighted interval automata" and "weighted deterministic register automata". Especially in the case of deterministic register automata, this model was only studied very recently (in a very difficult result Bojanczyk et al. (2021)) with respect to the problem of *language equivalence*, where the result cannot be applied to minimization. Our proof is a non-trivial analysis of the dynamics of such automata in producing minimum values.

**Organization:** In Section 2, we recall basic definitions and fix notation. Section 3 defines abductive/counterfactual explanations, and proves non-interpretability results. Positive results are in Section 4. We conclude in Section 5. Some proofs/details are in Supplementary Materials.

## 2 PRELIMINARIES

**Basic Notation:** Let $\Sigma$ be a set (called the *alphabet*), where the elements in our sequences range. In this paper, we consider sequences of rationals, i.e., $\Sigma \subseteq \mathbb{Q}$. We write $\Sigma^*$ to denote the set of finite sequences, whose elements range over $\Sigma$. A *language* over $\Sigma$ is simply a subset of $\Sigma^*$. We use $\overline{L}$ to denote the complement of $L$, i.e., $\Sigma^* \setminus L$. The empty sequence is denoted by $\varepsilon$. The length of the sequence $s \in \Sigma^*$ is denoted by $|s|$.

**Binary Classifiers for Sequences:** We study sequential models as representations of binary classifiers for $\Sigma^*$. Any binary sequence classifier $f : \Sigma^* \to \{0, 1\}$ can naturally be identified with the language $L_f = \{w \in \Sigma^* : f(w) = 1\}$. We interchangably use notation $L_f$ and $L(f)$ in the paper.

**Dynamic Time Warping:** Next, we discuss a few distance measures over sequences of rational numbers. The Dynamic Time Warping (DTW) Nielsen (2019) is a similarity measure between two non-empty sequences of possibly different lengths $v = v_1, ..., v_n$ and $w = w_1, ..., w_m$. In particular, DTW tries to capture the intuition that the same time series displayed at different speed should be similar. For example, it is the case that $d_{\text{DTW}}(v, w) = 0$, whenever $v = 0$ and $w = 0, 0, 0, 0$. In this sense, $d_{\text{DTW}}$ does not satisfy triangle inequality.

More precisely, the DTW function $d_{\text{DTW}} : \Sigma^+ \times \Sigma^+ \to \mathbb{Q}$ depends on a distance function $c : \Sigma \times \Sigma \to \mathbb{Q}$. We employ natural distance functions for $\Sigma = \mathbb{Q}$: $c(a, b) = |a - b|$ and the discrete metric (i.e. $c(a, b) = 1$ if $a \neq b$, and $c(a, b) = 0$ if $a = b$). We then define $d_{\text{DTW}}$ as:

$$d_{\text{DTW}}(v, w) := \min_P \sum_{(i,j) \in P} c(v_i, w_j)$$

where $P = ((i_0, j_0), \ldots, (i_\ell, j_\ell))$ is a sequence of matches between $v$ and $w$ satisfying: (1) $(i_0, j_0) = (1, 1)$, (2) $(i_\ell, j_\ell) = (n, m)$, and (3) for all $0 \leq k < \ell$, $(i_{k+1}, j_{k+1}) \in \{(i_k + 1, j_k), (i_k, j_k + 1), (i_k + 1, j_k + 1)\}$. The minimum is taken over all possible sequences, that satisfy these requirements. The idea is that similar sections of $v$ and $w$ will be matched together by $P$. One element of $v$ can be matched to multiple elements of $w$ and vice versa. When talking about time series, this corresponds to "warping" the time of one series to be more similar to the other.

**Hamming distance:** Hamming distance is a measure of difference between two sequences $v = v_1, \ldots, v_n$ and $w = w_1, \ldots, w_n$ of equal length. It is the number of positions at which corresponding letters of $v$ and $w$ differ. We can define the Hamming distance $d_H$ as:

$$d_H(v, w) := |\{i \in \{1, \ldots, n\} : v_i \neq w_i\}|$$

In other words, it is the minimum number of substitutions required to change one sequence to another.

**Edit distance:** The edit distance between two sequence $v$ and $w$ is the minimum number of operations required to transform one sequence $v$ to another sequence $w$ by using insertion, deletion and substitution of a single letter. Formally,

$$d_E(v, w) = \min_P \left( \sum_{(i,j) \in P} c(v_i, w_j) \right) + c_{\text{insdel}}(n + m - 2|P|),$$

where $P = ((i_0, j_0), \ldots, (i_\ell, j_\ell))$ is a *monotone* sequence of matches, i.e., for any $k < k'$ we have $i_k < i_{k'}$ and $j_k < j_{k'}$. In this definition, every matched pair $(i, j)$ contributes the matching/substitution cost $c(v_i, w_j)$ and the $n + m - 2|P|$ unmatched symbols from $v$ and $w$ incur some constant cost $c_{\text{insdel}}$. For finite alphabets $\Sigma$, we will consider the usual choice $c(a, a) = 0$, $c(a, b) = 1$ for $a \neq b$ and $c_{\text{insdel}} = 1$, unless specified otherwise.

**Complexity classes and complexity measures:** We assume basic familiarity with computability and complexity theory (e.g. see Sipser (1997)). In particular, we use standard complexity classes like NP (the set of problems solvable in nondeterministic polynomial-time), and PSPACE (the set of problems solvable in deterministic polynomial-space). Moreover, it is the case that NP $\subseteq$ PSPACE, and that complete problems for these classes are widely believed not to admit polynomial-time solutions. We will represent a rational number as $a/b$ with $a, b \in \mathbb{Z}$. As usual, numbers are almost always represented in binary. We provide here the following exception to this rule. A counterfactual query takes a distance upper bound $k \in \mathbb{N}$ as part of the input so that only $w \in \Sigma^*$ of distance at most $k$ is permitted as a counterfactual. We will represent $k$ in *unary* because one is typically interested in counterfactuals that lies in the proximity of a given sequence $v$ (i.e. of a small distance away from $v$).

## 3 ABDUCTIVE AND COUNTERFACTUAL EXPLANATIONS

We first formalize the notions of abductive and counterfactual explanations sequential models. These notions were defined for sequential models (e.g. see Marques-Silva & Ignatiev (2023); Barceló et al. (2020)). We then mention negative results regarding generating such explanations for the popular sequential models of RNN and Transformers.

**Abductive explanations:** Given subsets $F_1, \ldots, F_n \subseteq \Sigma^*$, which we call "features", an *abductive explanation* such that $M(v)$ for a model $M$ and $v \in \Sigma^*$ is a (subset-)minimal subset $S \subseteq \{F_1, \ldots, F_n\}$ such that $v$ satisfy all features (i.e. $v \in \bigcap_{X \in S} X$), that $M(w) = M(v)$ for any sequence $w$ satisfying all features in $S$ (i.e. $w \in \bigcap_{X \in S}$). [Note that, by convention, $\bigcap_{X \in S} X = \Sigma^*$, whenever $S = \emptyset$.] For a finite alphabet $\Sigma$, one can use regular languages over $\Sigma$ to represent $F_1, \ldots, F_n$.

**Example 1.** We provide a small example to illustrate the concept of abductive explanations. In a simple document classification task, a feature could refer to the existence of a certain string (e.g. "Republicans" or "Democrats" or "Election"). These can be captured using regular languages, e.g.,

$$L_1 = \Sigma^* (\text{R+r}) \text{epublicans} \Sigma^* \qquad L_3 = \Sigma^* (\text{E+e}) \text{election} \Sigma^*$$
$$L_2 = \Sigma^* (\text{D+d}) \text{emocrats} \Sigma^* \qquad L_4 = \Sigma^* (\text{I+i}) \text{minnent} \Sigma^*$$

Suppose that we have trained a DFA $A$ to recognize documents about presidential election in the states. Suppose that $T$ is a text, whereby $T$ is in $L_1 \cap L_2 \cap L_3 L_4$. An abductive explanation could be $L_1, L_2, L_3$ since the words "imminent" and "Imminent" turn out not to be important in determining whether a text concerns US election, according to $A$. $\qquad\qquad\square$

We remark that using regular languages as features have various advantages. Firstly, they are sufficiently expressive in capturing concepts in a variety of applications. Secondly, a regular language representing a specific feature can itself be learned using an automata learning algorithm (e.g. Gold (1978); Angluin (1987a)). For example, suppose that we have trained a DFA $A$ that classifies whether an input email is a spam. Features could be: (1) a DFA $B$ that identifies if the email sender asks for money, and (2) a DFA $C$ identifying if there are dangerous links (a list of known dangerous URLs). It is likely that $B$ is a very complicated regular language, which is why an automata learning algorithm that can infer $B$ from examples can come in handy.

**Counterfactual explanations:**  Given a model $M$, a sequence $v$, and a number[2] $k$, a counterfactual explanation for $M(v) \in \{0, 1\}$ is a sequence $w$ such that $M(v) \neq M(w)$ and that the "distance" between $v$ and $w$ is at most $k$. Here, "distance" may be instantiated by Hamming distance, edit distance, or DTW. We provide examples of counterfactual queries in Section 4.

**Negative results:**  RNN Rumelhart et al. (1985) and Transformers Vaswani et al. (2017) are popular sequential models in the machine learning literature. RNN and Transformers are extensions of standard multi-layer perceptrons, so that input sequences of unbounded lengths can be processed by the neural network. To achieve this, RNN allows edges that feed back to the input neurons, while Transformers use self-attention mechanisms and positional encoding. Both models are known to be Turing-complete, as was proven by Siegelmann & Sontag (1995) and by Pérez et al. (2021). Here, it suffices to use finite alphabets $\Sigma$. Using these results, we can prove that generating abductive and counterfactual explanations is computationally difficult for RNN and Transformers.

**Proposition 1.** *The problem of checking the existence of an abductive explanation for RNN and Transformers are undecidable. The problem of checking the existence of a counterfactual explanation with respect to Hamming and Edit distance (resp. DTW) is NP-complete (resp. undecidable).*

We relegate the full proof of this proposition to the appendix. Intuitively, to prove the result, one first observes that abductive and counterfactual explanations can be used to encode universality/non-emptiness of a language (i.e. whether $L = \Sigma^*$ or $L \neq \emptyset$). The undecidability result can then be proven by using Turing-completeness of RNN and Transformers. To obtain NP-hardness, we use the result that counterfactual explainability is NP-hard for multi-layer perceptrons Barceló et al. (2020), which is then inherited by the more general model of RNN and Transformers.

Unfortunately, it is easy to show that abductive explanability is computationally intractable even for DFA, unless the number of features is fixed. See Appendix B for proof.

**Proposition 2.** *The problem of checking the existence of an abductive explanation for DFA is PSPACE-hard. When the number of features is fixed, the problem is solvable in polynomial-time.*

## 4    INTERPRETABLE MODELS FOR COUNTERFACTUAL QUERIES

It is possible to show that generating counterfactual explanations for DFA is polynomial-time solvable. Instead of first proving this result, and proving the more general results for deterministic interval automata and deterministic register automata (with a fixed number of registers), we directly prove this polynomial-time solvability for the more general models.

### 4.1    INTERVAL AUTOMATA

We will define the notion of interval automata. This setting is equivalent to an instantiation of the framework of *automata modulo theories* D'Antoni & Veanes (2021) over SMT-Algebra modulo Real Arithmetic. Intuitively, an interval automaton is just an automaton, where a transition may take an interval with rational endpoints. More precisely, let $\Psi$ be the set of intervals with rational

---

[2]Represented in unary, i.e., as the string $1^k$

end points (or infinity), i.e., $[l_i, u_i], [l_i, u_i), [l_i, \infty), (l_i, u_i], (-\infty, u_i], (l_i, u_i),$ or $(-\infty, \infty)$, where $l_i \leq u_i$ and $l_i, u_i \in \mathbb{Q}$. An *interval automaton* $\mathcal{A} = (\Sigma, Q, \Delta, q_0, F)$, where $\Sigma = \mathbb{Q}$, $Q$ is a finite set of control states, the transition relation $\Delta$ is a finite subset of $Q \times \Psi \times Q$, $q_0 \in Q$ is an initial state, and $F \subseteq Q$ is a set of final states. Given a sequence $w = a_1 \cdots a_n \in \mathbb{Q}^*$, a *run* of $\mathcal{A}$ on $w$ is a function $\pi : \{0, \ldots, n\} \to Q$ such that $\pi(0) = q_0$ and, for each $i \in \{0, \ldots, n-1\}$, we have $(\pi(i), P, \pi(i+1)) \in \Delta$ and $a_{i+1} \in P$. This run is said to be *accepting* if $\pi(n) \in F$, and we say that $\mathcal{A}$ accepts $w$ if there is an accepting run of $\mathcal{A}$ on $w$. The language $L(\mathcal{A})$ accepted by $\mathcal{A}$ is defined to be the set of strings $w \in \Sigma^*$ that are accepted by $\mathcal{A}$. We say that $\mathcal{A}$ is *deterministic* if there are no two distinct transitions $(p, P_1, q_1), (p, P_2, q_2) \in \Delta$ such that $P_1 \cap P_2 \neq \emptyset$. It is known that interval automata are determinizable D'Antoni & Veanes (2021).

**Example 2.** A simple example of a language that can be recognized by an interval automaton is a sequence of numbers in the range of $300$ and $400$. This can be captured by a single deterministic interval automaton with state $q$ (both initial and final) with a single transition $(q, [300, 400], q)$. Such a language could be of interests in modelling that a stock price is in a certain "trading range" (i.e. with no clear up/down trends). Consider an extension of this language, a trading strategy, where "buy" is recommended, when the closing prices are in a window between \$300 and \$400 for the past four data points, and the last recorded price is between \$300 and \$303, meaning within $1\%$ of the window's lower cut-off. Such a strategy can easily be represented using an interval automaton, whose language is the set of all sequences that get a "buy" recommendation. Consider the sequence $298, 305, 301, 320, 315, 302$, the strategy would recommend "buy", since the last four values are between $300$ and $400$, and the last value is between $300$ and $303$. A counterfactual explanation for this sequence on a model representing this strategy, would be the sequence $298, 305, 299, 320, 315, 302$, which would not get a "buy" recommendation, since its third value is now under $300$.

**Remark.** The standard definition of finite automata (e.g. see Sipser (1997)) over finite alphabet can be obtained from the above definition by allowing intervals of the form $[i, i]$.

**Theorem 1.** *The complexity of a counterfactual query with respect to a deterministic interval automaton is polynomial-time solvable.*

We sketch the result for DTW distance measure $d$; the proof for the rest is similar and is given in supplementary materials. *Here, we deal with a decision problem, but later we remark how a counterfactual of a small DTW-distance can be easily extracted from our algorithm.* We are given a function $f : \Sigma^* \to \{0, 1\}$ given by the DFA $A$, an input sequence $v \in \Sigma^*$, and $k \in \mathbb{N}$. We will show that checking the existence of $w \in \Sigma^*$ such that $d(v, w) \leq k$ and $f(v) \neq f(w)$ can be done in PTIME. Note first that the value $f(v)$ can be easily computed in PTIME.

Without loss of generality, we assume that $f(v) = 1$. The case of $f(v) = 0$ is similar. We first construct a DFA $B := \bar{\mathcal{A}} = (\Sigma, Q, \Delta, q_0, F)$ for the complement language of $L(\mathcal{A})$. This can be achieved by first "completing" the transition relation and then swapping final/non-final states. For example, suppose that the available transitions in $\mathcal{A}$ from $q$ are $(q, [1, 2], p_1)$ and $(q, [4, 5], p_2)$. Completing here means that we add a "dead-end" state $q_r$ and add transitions $(q_r, (\infty, \infty), q_r)$ and $(q, P, q_r)$ for each $P \in \{(-\infty, 1), (2, 4), (5, \infty)\}$. The computation of $B$ can be done in PTIME.

We now construct a weighted graph $G = (V, E, S, T)$, where $V$ is a set of vertices, $E \subseteq V \times \mathbb{Q}_{\geq 0} \times V$ is a finite set of weighted edges, and $S, T \subseteq V$ are sets of start/target vertices. Furthermore, it is the case that the minimum weight $W \in \mathbb{Q}_{\geq 0}$ of paths from $S$ to $T$ in $G$ corresponds $d(v, w)$ where $w \in L(B)$, such that $d(v, w)$ is minimized. Let $v = a_1 \cdots a_n$. Here, weight of a path refers to the sum of the weights of the edges in the path. Polynomial-time upper bound follows since computing $W$ can be solved using a polynomial-time shortest-path algorithm.

To simplify our presentation, we assume that each transition $(p, P, q) \in \Delta$ is associated with a closed interval $P = [r, s]$. The case when $P$ is not closed interval (e.g. $(1, 7]$) can be taken care of easily by adding a flag in $G$, so as to indicate whether $W$ is the infimum or can be achieved. See supplementary materials. Define $V = Q \times \{1, \ldots, n\}$, $S = \{q_0\} \times \{1\}$, and $T = F \times \{n\}$. For each transition $(p, P, q) \in \Delta$, we add the following edges:

1. $(p, i)$ to $(q, i)$ with weight $\inf_{r \in P} |r - a_i|$. This means that we match $a_i$ with the current value in the counterfactual string.

2. $(p, i)$ to $(q, j)$ for every $j \in \{i+1, \ldots, n\}$ with weight $\inf_{t \in P} \sum_{h=i}^{j-1} |t - a_h|$. This means that we match the current value $t$ in the counterfactual string with $a_i, \ldots, a_{j-1}$.

The weight in the last item can be computed in polynomial time. For example, we can observe that this is a linear program optimization with a sum of absolute values as in the objective function: minimize $\sum_{i=1}^{n-1} |x_i|$ with $x_i = t - a_i$ for each $i$ and $r \leq t \leq s$. We may then use a standard trick to replace each $|x_i|$ by $x_i^+ + x_i^-$, for new variables $x_i^+$ and $x_i^-$, and replace $x_i$ by $x_i^+ - x_i^-$. Finally, we assert that $x_i^+, x_i^- \geq 0$.

**Remark.** While we calculate these weights, we can also label each edge with the optimal value for $r$ (for edges created in the first item) or $t$ (for edges created in the second item). When we then execute a shortest path algorithm, we can have it return the shortest path to us. The sequence of labels we see when we follow this path, will be the counterfactual explanation.

## 4.2 EXTENSIONS

**Adding registers:**  We consider the extension of DFA with $k$ registers $r_1, \ldots, r_k$, resulting in the model $k$-DRA of *deterministic $k$-register automata* (e.g. Moerman et al. (2017); Cassel et al. (2016)). We will show that counterfactual queries for such $k$-DRA can still be answered in PTIME.

A good example of a language recognized by $k$-DRA (in fact 1-DRA) is the set of non-decreasing sequences greater than some value $a$; it can be recognized by remembering the last seen value in the register and then comparing it with the next value in the input.

To define $k$-DRA we first define nondeterministic $k$-register automata ($k$-RA). More formally, a $k$-RA $\mathcal{A} = (Q, \Delta, q_0, F)$ consists of a finite set $Q$ of states, the initial state $q_0 \in Q$, the set $F$ of accepting states, and a finite transition relation $\Delta$, with each transition in it of the form $(p, \varphi, \psi, q)$, where $\varphi(\bar{r}, curr)$ is a *guard* — comparing the values of the registers and the currently read value in the input — and $\psi(\bar{r}, curr)$ is a set of *assignments* for updating the registers $\bar{r} = r_1, ..., r_k$. More precisely, a guard is simply a set of (in)equalities of the form $r_i \sim curr$ or $r_i \sim r_j$, where $\sim \in \{=, \leq, \geq, <, >\}$. In addition, an assignment is of the form $r_i := c$ (for a constant $c \in \mathbb{Q}$), $r_i := r_j$, or $r_i := curr$. Inside $\psi$, we only allow *at most one assignment* $r_i := ...$ for each $1 \leq i \leq k$. We next define the notion of runs and accepting runs. A *configuration* is a pair $(p, \mu)$, where $p \in Q$ and $\mu : \{r_1, \ldots, r_k\} \to \mathbb{Q}$. An *initial configuration* is of the form $(q_0, \bar{0})$, where $\bar{0}$ assigns 0 to each $r_i$. Given a sequence $w = a_1 \cdots a_n \in \mathbb{Q}^*$, a run on $w$ is a mapping $\pi$ of each $i \in \{0, \ldots, n\}$ to a configuration $\pi(i) := (q_i, \mu_i)$ such that $\pi(0)$ is an initial configuration, and for each $l = 0, \ldots, n-1$ there exists a transition $(q_l, \varphi, \psi, q_{l+1}) \in \Delta$ such that $\varphi(\mu_l(\bar{r}), a_{l+1})$ is satisfied and that the following holds true for $\mu_{l+1}$ with $1 \leq i \leq k$:

- $\mu_{l+1}(r_i) = \mu_l(r_i)$, when no assignment of the form $r_i := ...$ appears in $\psi$
- $\mu_{l+1}(r_i) = \mu_l(r_j)$, when an assignment of the form $r_i := r_j$ appears in $\psi$
- $\mu_{l+1}(r_i) = c$, when an assignment of the form $r_i := c$ with $c \in \mathbb{Q}$ appears in $\psi$
- $\mu_{l+1}(r_i) = a_{l+1}$, when an assignment of the form $r_i := curr$ appears in $\psi$

We say that the run $\pi$ is *accepting* if the last state $q_n$ is a final state. The language accepted by $\mathcal{A}$ contains all sequences $w \in \mathbb{Q}^*$, on which there is an accepting run of $\mathcal{A}$.

Having defined $k$-RA, we may now impose determinism by allowing at most one transition for each value $a$ seen in the input. More precisely, we say that a $k$-RA $\mathcal{A} = (Q, \Delta, q_0, F)$ is *deterministic* if it is not the case that there are two different transitions $(p, \varphi_1, \psi_1, q), (p, \varphi_2, \psi_2, q') \in \Delta$ such that $\varphi_1(\mu(\bar{r}), a)$ and $\varphi_2(\mu(\bar{r}), a)$ are both true for some $a \in \mathbb{Q}$, and $\mu : \{r_1, \ldots, r_k\} \to \mathbb{Q}$. Checking whether a $k$-RA is deterministic can be done in PTIME; in particular, this is a simple instance of a linear programming, which is PTIME solvable.

As we mentioned above, $k$-DRA can be used to recognize the language of non-decreasing sequences in $\mathbb{Q}^+$ all of whose elements are greater than some value $a$.

**Example 3.** For another example of where $k$-DRA can be useful, we look again at financial data. Financial data is often annotated, using relatively straightforward criteria. One such annotation is the notion of an "uptrend". We can define a sequence to be an uptrend, if the values of the local maxima are monotonically increasing and the values of the local minima are monotonically increasing. For this example, we assume that the data is "smooth enough", which can be achieved through pre-processing steps, so that we do not mistakenly consider minor fluctuations as local minima or maxima. A register automaton with two registers can easily decide if a sequence is an uptrend, two registers are used to

save the previous high and the previous low. Let us consider the sequence $1, 5, 3, 5, 7, 9, 6, 8, 10, 12$, which has the local maxima $5, 9$ and the local minima $3, 6$ (we do not count the first value as a local minimum or last value as a local maximum). Since both are monotonically increasing, this sequence represents an uptrend. A counterfactual explanation for this sequence on this model would be $1, 5, 3, 5, 7, 9, 2, 8, 10, 12$, where the six has been changed to a two. Since the new local minima $3, 2$ are no longer monotonically increasing, this sequence is not an uptrend.

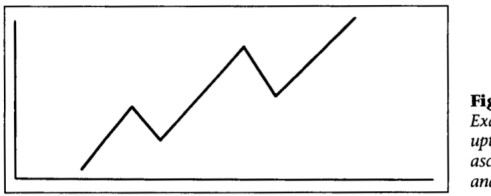

**Figure 4.1a**
*Example of an uptrend with ascending peaks and troughs.*

Figure 1: Graph of an uptrend for example 3, taken from Murphy (1999)

**Theorem 2.** *The complexity of evaluating a counterfactual query with respect to $k$-DRA is in PTIME for Hamming distance, edit distance, and DTW assuming a fixed $k$.*

*Proof.* To proof this we construct a weighted graph, just like in the proof of Theorem 1. *We deal with the decision version of the problem, but a counterfactual of a small distance can be easily computed from our algorithm just like in the case of Theorem 1.* The immediate problem here adapting this idea is that the number of configurations of a $k$-DRA is infinite. For this reason, we restrict our search for a counterfactual to words over the values that appear as constants in the $k$-DRA and the values present in the input word. This creates a finite set of configurations whose size is exponential only in $k$. Hence, when we fix $k$, we will solve a shortest path problem on a polynomial-sized graph, just as in the case for DFA.

We will consider only the case that each guard contains only a set of weak inequalities, i.e., of the form $x \sim y$, where $x, y$ are registers (i.e. of the form $r_i$), constants (i.e. concrete values in $\mathbb{Q}$), or the current value $curr$ and $\sim \in \{\leq, \geq\}$. We can handle strict inequality in a similar way as in the proof of Theorem 7. Let $v = a_1 \cdots a_n \in \mathbb{Q}^+$. Without loss of generality, assume $v \notin L(\mathcal{A})$. In the following, we want to find $w \in \mathbb{Q}^+$ such that $w \in L(\mathcal{A})$ and that $d(v, w)$ is minimized. We will assume that $d$ is the DTW distance measure with $c(x, y) = |x - y|$ (the proof for discrete metric is the same), to adapt the proof to Hamming and edit distance, the same modifications as for Theorem 1 can be applied. We begin with a lemma, which we will use to restrict the set of configurations.

**Lemma 3.** Let $D$ be the finite set of constants appearing in $\mathcal{A}$ or in $v$. For each $w \in \mathbb{Q}^+$ with $w \in L(\mathcal{A})$, there exists $u \in D^+$ such that $d(v, u) \leq d(v, w)$ and $u \in L(\mathcal{A})$.

*Proof.* Let $w = b_1 \cdots b_m$. Let $\pi$ be an accepting run of $w$ on $\mathcal{A}$, with $(q_i, \mu_i) := \pi(i)$ for $0 \leq i \leq m$. Let $(q_i, \varphi_i, \psi_i, q_{i+1})$ be the transition between $\pi(i)$ and $\pi(i+1)$ for $0 \leq i \leq m - 1$.

For each $i \in \{1, \ldots, m\}$, select $l \in \{1, \ldots, n\}$ such that $\sum_{j \in M} c(a_j, a_l)$ is minimal, where $M = \{j | (i, j)$ is a match when calculating $d(v, w)\}$. We then set $u_i := a_l$ if $\varphi_i(\mu_i(\bar{r}), a_l)$ is satisfied. If $a_l$ is too large, we instead set $u_i$ to the smallest upper bound on $curr$ in $\varphi_i$ for register values of $\mu_i$, and if $a_l$ is too small, we instead set $u_i$ to the largest lower bound. This guarantees that $\pi$ is also an accepting run of $u := u_1, \ldots, u_m$ on $\mathcal{A}$, meaning $u \in L(\mathcal{A})$. Furthermore, for each match $(i, j)$ in the matching sequence of $d(v, w)$, we have $c(v_i, u_j) \leq c(v_i, w_j)$, meaning $d(v, u) \leq d(v, w)$. $\square$

This lemma essentially says that it suffices to restrict ourselves to values in the automaton $\mathcal{A}$ or the string $v$ when finding a counterfactual. Firstly, since $\mathcal{A}$ starts with $0$ in all registers and we have an assignment for each register in each transition (otherwise, stays the same), it suffices to restrict each register value to $D$. As before, we can construct a graph $G = (V, E, S, T)$, but this time the set $V$ of vertices is $Q \times \{1, \ldots, n\} \times D^k$. Here, in each tuple $(q, i, j_1, \ldots, j_k) \in V$ the value $j_l$ records which of the values in $D$ is stored in the $l$th register. We define $S = q_0 \times \{1\} \times \{0\}^k$, and $T = F \times \{n\} \times D^k$. The edge relation is defined in the same way as in the proof of Theorem 1; For each $(p, \varphi, \psi, q) \in \Delta$ we add to $E$ the edges:

1. $(p, i, \theta)$ to $(q, i, \theta[\psi(\theta, curr)])$ with weight $\min_{z \in D \text{ s.t. } \varphi(\theta, z)} c(z, a_i)$ for every $i \in \{1, \dots, n\}$, every $\theta \in D^k$, and every $curr \in D$, where $\theta[\psi(\theta, curr)]$ are the values of the registers after $\psi$ has been applied on the values $\theta$.

2. $(p, i, \theta)$ to $(q, j, \theta[\psi(\theta, curr)])$ with weight $\min_{z \in D \text{ s.t. } \varphi(\theta, z)} \sum_{h=i}^{j-1} c(z, a_h)$ for every $i \in \{1, \dots, n\}$, $j \in \{i + 1, \dots, n\}$, $\theta \in D^k$, and $curr \in D$.

in particular, this can be done because we have a finite set $D$ of values. Note that now $G$ is polynomial, except in the number $k$ of registers. As for Theorem 1, we can apply a shortest path algorithm on $G$, to get $\min_{u \in \mathbb{Q}^+} d(v, u)$ as the weight of a shortest path. With this the counterfactual query can be answered. $\qquad\square$

**Adding nondeterminism:**   We conclude with an observation that the extension of DFA with nondeterminism (i.e. NFA) is intractable with respect to counterfactual queries.

**Proposition 3.** *The problem of checking the existence of counterfactuals for NFA with respect to Hamming and edit distance is NP-hard, while the same problem is PSPACE-hard for DTW.*

## 5   Conclusions, Limitations, and Future Work

**Summary and Discussion:**   We have seen that popular sequential models like RNN and Transformers are either difficult to interpret with respect to two types of explanations in FXAI, i.e., abductive and counterfactual explanations. For sequences over a finite alphabet, it has been hinted by the numerous available works on automata extraction from RNN (e.g. Dong et al. (2020); Weiss et al. (2018); Okudono et al. (2020); Wang et al. (2017); Jacobsson (2005); Bollig et al. (2022)) that Deterministic Finite Automata (DFA) might be more interpretable. Interestingly, we showed that DFA is intractable (PSPACE-complete) for abductive explanations with respect to features represented as regular languages. On the positive side, we provide a polynomial-time algorithms for computing counterfactual explanations for extensions of DFA to infinite alphabets — including deterministic interval automata and deterministic $k$-register automata (for fixed $k$).

Combined with existing automata learning algorithms (e.g. Gold (1978); Angluin (1987a); Moerman et al. (2017); Drews & D'Antoni (2017); Argyros & D'Antoni (2018)), our polynomial-time algorithms could potentially be used in FXAI as follows: (1) learn an automaton (either from datasets, or from more complex models like RNN, e.g., see Weiss et al. (2018); Okudono et al. (2020)), (2) use our PTIME algorithm to generate a counterfactual explanation with respect to a specific input $w$.

Many theoretical papers on RNN and Transformers (e.g. Siegelmann & Sontag (1995); Pérez et al. (2021)) assume rational numbers of *arbitrary precision*, which is also what we assume here. In practice, it is often the case that rational numbers of a *fixed precision* (i.e. floating point numbers with a fixed number of bits) are employed. With a fixed precision, models like RNN and Transformers have only finite memory, which are then no more powerful than DFA. However, when the number of bits permitted is allowed as a parameter (i.e. finite but not fixed precision), then the amount of memory is still exponential in the number of bits. We leave it for future work to study the complexity of counterfactual queries of such RNN and Transformer models.

Our paper deals with explainability in FXAI. This type of explainability falls within the category of *post-hoc local model-agnostic explanations*. This represents only a small subset of the landscape of XAI Molnar (2022); Gunning et al. (2019); Linardatos et al. (2021); Belle & Papantonis (2021), where explanations may involve subjective/human elements (e.g. visual explanation).

**Future work:**   We propose three future research directions. Firstly, can we restrict RNN and Transformers, so as to make the models interpretable? One potential candidate is to permit only Transformer encoders. In fact, it was shown (e.g. Hahn et al. (2021); Hao et al. (2022); Yao et al. (2021)) that some regular languages cannot be recognized by Transformers encoders. Secondly, is it possible to come up with subsets of "regular" features, where polynomial-time solvability can be recovered for DFA? Thirdly, our result indicates that $k$-DRA is an interpretable model, for fixed $k$. Can we extend the work of Weiss et al. (2018); Okudono et al. (2020) on DFA extraction from RNN using DFA learning algorithm to the $k$-DRA model? In particular, how do we exploit learning algorithms for $k$-DRA Howar et al. (2012); Cassel et al. (2016); Moerman et al. (2017)?

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

## APPENDIX

## A    PROOF SKETCHES FOR PROPOSITION 1

We first recall the formal definition of RNN (see Chen et al. (2018) and (Siegelmann & Sontag, 1995)):

**Definition A.1.** We define an *arbitrary precision RNN binary classifier $R$* with a *single recurrent layer* and $k$ *feed forward layers* as a tuple $\langle \Sigma, N, h_{-1}, W, W', \sigma, W'', b, f, k \rangle$, in which

- $\Sigma$ is a finite *input alphabet*,
- $N$ is a finite set of *neurons*,
- $h_{-1} \in \mathbb{Q}^{|N|}$ is the *initial activation vector*,
- $W \in \mathbb{Q}^{|N| \times |N|}$ is the *transition matrix* for the recurrent part,
- $W' = (W'_a)_{a \in \Sigma_\$}$ is a $\Sigma_\$$-indexed family of *bias vectors*, with $W'_a \in \mathbb{Q}^{|N|}$, where \$ is a fresh delimiter symbol and $\Sigma_\$ := \Sigma \cup \{\$\}$,
- $\sigma$ is the *activation function* of the recurrent part,
- $W'' = W''^{(1)}, W''^{(2)}..., W''^{(k)}$ is a sequence of *weight matrices* for the feed forward part, with $W''^{(i)} \in Q^{D_i \times D_{i+1}}$ for some sequence of dimensions $D_0, D_1, ..., D_k$ where $D_0 = |N|$ and $D_k = 1$,
- $b = b^{(1)}, b^{(2)}, ..., b^{(k)}$ is a sequence of *bias vectors*, with $b_i \in Q^{D_i}$,
- $f = f^{(1)}, f^{(2)}, ..., f^{(k)}$ is a sequence of *activation functions*,
- $k$ is the number of *feed forward layers* .

We define the *hidden state vector* $h_{s,t} \in \mathbb{Q}^{|N|}$ after reading the first $t - 1$ letters of an input sequence $s$, with $h_{s,t} := \sigma(W \cdot h_{s,t-1} + W'_{s_t})$ where $h_{s,-1} := h_{-1}$.

It suffices to show hardness only for finite alphabets $\Sigma$. To this end, we define the classification of a string $s \in \Sigma^*$ by the RNN classifier as:

$$R(s) = f^{(k)}(...f^{(1)}(W''^{(1)} \cdot h_{s,n} + b^{(1)})...)$$

*Proof sketch of RNN/Counterfactual Part of Proposition 1.* We sketch the proof first only for counterfactual queries with respect to DTW. For a proof by contradiction, we assume that checking the existence of a counterfactual explanation is decidable for an RNN binary classifier with a single layer feed forward part.

Let $M$ be an arbitrary Turing machine, that does not halt after 0 or 1 steps on the empty input. Theorem 2 of Siegelmann & Sontag (1995) (in the notation of Chen et al. (2018)) states, that there exists an RNN over $\Sigma = \{a\}$, with a designated neuron $n$, such that $h_{s,t}(n) = 1$ if $M$ stops after $t$ steps for the empty input and $h_{s,t}(n) = 0$ otherwise. From this we can construct an RNN classifier $R = \langle \Sigma, N, h_{-1}, W, W', \sigma, W'', b, f, k \rangle$, by taking $\Sigma, N, h_{-1}, W, W'$ and $\sigma$ from the construction in that proof and setting $k := 1$, $W'' := W^{(1)} := e_n^T$ (where $e_n$ denotes the vector with a 1 in the $n$-th coordinate and 0's elsewhere), $b := b^{(1)} := 0$, and $f := f^{(1)} := id$ (where $id$ is the identity), such that there exists a string $s \in \Sigma^*$ with $R(s) = 1$, if and only if $M$ halts on the empty input. We now consider counterfactual queries on $R$, with respect to DTW. Given $k \in \mathbb{N}$, we set $v := a \in \Sigma^+$ meaning $R(v) = 0$. We are looking for a string $w \in \Sigma^+$ such that $R(w) = 1$ and $DTW(v, w) \leq k$. We note that on the alphabet $\Sigma = \{a\}$ we have $DTW(x, y) = 0$ for all $x, y \in \Sigma^+$. Therefore,

$DTW(v, w) = 0$ for all possible $w \in \Sigma^+$, meaning that answering this counterfactual query is equivalent to checking the emptiness of class $C = \{s \in \Sigma | R(s) = 1\}$. However, the class $C$ is non-empty if and only if $M$ halts. Answering the counterfactual query of any RNN, with respect to DTW, therefore decides the Halting Problem, which is known to be undecidable. $\square$

Next we sketch the proof for counterfactual explanations for RNN with respect to Hamming and edit distance. The full proof can be found in Appendix C. The full proof for Transformers can be found in Appendix D. The undecidability proof of abductive explanations is essentially the same as the proof of Proposition 2, except that we use RNN and Transformers to recognize the language $\{c\}$, and then the undecidability of non-emptiness of languages of RNN/Transformers Siegelmann & Sontag (1995); Pérez et al. (2021).

*Hamming distance:* NP-hard can be achieved by a reduction from the same problem for multilayer perceptrons Barceló et al. (2020). This problem is called *MinimumChangedRequired (MCR)* by the authors. We prove the NP hardness for Hamming and edit distance by considering an arbitrary MLP $M$ with $k$ layers, whose weight matrix, bias vector and activation function for each layer $i$ is denoted by $W''^{(i)}, b^{(i)}$ and $f^{(i)}$ respectively and with input from $\{0, 1\}^d$.

*Edit distance:* We construct an RNN binary classifier $R_M$ over $\Sigma = \{0, 1\}$, such that the language recognized by $R_M$ is exactly the language recognized by $M$. This $R_M$ is composed of two parts: (1) A single layer recurrent network (2) The feedforward network $M$ with $k$ layers. For any input sequence $w = w_1, w_2, ..., w_d$, the recurrent network of $R_M$ converts the input sequence into a vector $\begin{bmatrix} w_1 & w_2 & ... & w_d \end{bmatrix}^T$ using the upper shift matrix. The vector is then processed by $M$. Since the counterfactual query with regard to Hamming distance is NP complete for MLPs Barceló et al. (2020), it must also be NP-hard for $R_M$ with regard to Hamming distance. Thus, the complexity of evaluating counterfactual queries for RNN classifiers with regard to Hamming distance is NP-hard.

The NP-hardness proof for edit distance follows the same idea. We construct an RNN binary classifier $R'_M$ over $\Sigma' = \{0, 1, \$_1, \$_2, ..., \$_d\}$. Any input sequence $w$ is modified to $w' := \$_1, w_1, \$_2, w_2, ...\$_d, w_d$. We construct $R'_M$ such that a word is accepted iff every other symbol is the appropriate delimiter symbol and, the word without delimiter symbols would be accepted by $M$. If $v', w' \in \Sigma^*$ are of this form and have the same length then, $d_H(v, w) = d_E(v', w')$, because the shortest sequence of instructions which changes $v'$ to $w'$ consists of only substitutions of elements in $\Sigma$ with elements in $\Sigma$. After construction of $R'_M$ the rest of the proof follows the proof for Hamming distance. The precise construction of $R_M$ and $R'_M$ are in Appendix C.

# B  PROOF OF PROPOSITION 2

*Proof.* The proof is by a polynomial-time reduction from the non-emptiness of intersection of DFA $A_1, \ldots, A_n$, say, over an alphabet $\Sigma = \{a, b\}$. Let $c$ be a new letter not in $\Sigma$, and let $\Gamma = \{a, b, c\}$. Let $B$ be a DFA that recognizes the language $\{c\}$, and define a new DFA $A'_i$ that recognizes $L(A_i) \cup \{c\}$. Observe that $c \in L(B)$ and $c \in \bigcap_{i=1}^n L(A'_i)$. An abductive explanation requires us to argue that there is a non-empty subset $S \subseteq \{1, \ldots, n\}$ such that $\overline{L(B)} \cap \bigcap_{i \in S} L(A'_i) = \emptyset$. This is the same as saying that there is $S \subset \{1, \ldots, n\}$ such that $\bigcap_{i \in S} L(A_i) = \emptyset$, which amounts to saying that $\bigcap_{i=1}^n L(A_i) = \emptyset$. This completes the reduction.

For a PTIME upper bound with a fixed number $k$ of features, one can simply enumerate all subsets $S$ of these features (there are exponentially many in $k$, but this is a constant), and observe that the intersection of $m \leq k$ regular languages (i.e. the features) is polynomial in the size of the regular languages, but exponential in $k$ (which is a constant). $\square$

# C  RNN

**Definition C.1** (Adapted from Barceló et al. (2020)). We define an arbitrary precision multilayer perceptron (MLP) $M$ over $\{0, 1\}^d$ as a tuple $\langle k, W'', b, f \rangle$ in which,

- $k$ is the total number of layers,
- $W''$ is a sequence of weight matrices $W''^{(1)}, ..., W''^{(k)}$,

- $b$ is a sequence of bias vectors $b^{(1)}, ..., b^{(k)}$ and,
- $f$ is a sequence of activation functions $f^{(1)}, ..., f^{(k)}$ where $f^{(1)}, ..., f^{(k-1)}$ are the $ReLU$ function and $f^{(k)}$ is the $step$ function, applied elementwise, where,

$$ReLU(x) := max(0, x) \text{ and } step(x) := \begin{cases} 0 & \text{if x} < 0 \\ 1 & \text{otherwise} \end{cases}$$

The output of $M$ on $w \in \{0, 1\}^d$ is defined as:

$$M(w) := f^{(k)}(W''^{(k)}\dot{(}...f^{(1)}(W''^{(1)} \cdot w + b^{(1)})...) + b^{(k)})$$

## C.1 PROOF OF PROPOSITION 1 FOR RNN

*Proof. (1)Hamming/(2)edit distance:* In NP: For a given input $v \in \Sigma^*$, RNN classifier $R$ and $k \in \mathbb{N}$, non-deterministically guess $w$ such that $R(w) \neq R(v)$ and $d_H(v, w) \leq k$ for Hamming or $d_E(v, w) \leq k$ for edit distance. For edit distance, also guess a sequence of $k$ or fewer instructions (substitute, insert, delete), such that applying them to $v$ yields $w$. Because Hamming distance, edit distance and $R(v), R(w)$ can be verified/computed in polynomial time, the problems of answering counterfactual queries for RNN with regard to Hamming and edit distance are in NP.

NP-hard: For Hamming distance, we construct an RNN binary classifier $R_M := \langle \Sigma, N, h_{-1}, W, W', \sigma, W'', b, f, k \rangle$ where,

- $\Sigma := \{0, 1\}$,
- $N := \{1, ..., d\}$,
- $h_{-1} \in \mathbb{Q}^d$ is the zero vector,
- $W'_0, W'_1 \in \mathbb{Q}^d$, where $W'_0$ is the zero vector, $W'_1 := [0 \ ... \ 0 \ 1]^T$,
- $W \in \mathbb{Q}^{d \times d}$ is the upper shift matrix: $\begin{bmatrix} 0 & 1 & 0 & ... & 0 \\ 0 & 0 & 1 & ... & 0 \\ & & . & & \\ & & . & & \\ & & . & & \\ 0 & 0 & 0 & ... & 1 \\ 0 & 0 & 0 & ... & 0 \end{bmatrix}$
- $\sigma$ is the identity function,
- $W'', b, f, k$ are the same as in $M$.

For $w = [w_1 \quad w_2 \quad ... \quad w_d]^T \in \Sigma^d$, we write $R_M(w) := R_M(w_1.w_2...w_d)$. Because of the choice of $W$ and $W'$, the hidden state vector at time step $d$ is:

$$h_{w,d} = \sigma(...\sigma(W \cdot \sigma(W \cdot \sigma(W \cdot h_{-1} + W'_{w_1}) + W'_{w_2}) + W'_{w_3})...)$$

$$= \sigma\left(...\sigma\left(W \cdot \sigma\left(W \cdot \sigma\left(W \cdot [0 \ 0 \ ... \ 0]^T + \begin{bmatrix} 0 \\ 0 \\ . \\ . \\ . \\ 0 \\ w_1 \end{bmatrix}\right) + \begin{bmatrix} 0 \\ 0 \\ . \\ . \\ . \\ 0 \\ w_2 \end{bmatrix}\right) + \begin{bmatrix} 0 \\ 0 \\ . \\ . \\ . \\ 0 \\ w_3 \end{bmatrix}\right)...\right)$$

$$= \sigma \left( ...\sigma \left( W \cdot \sigma \left( W \cdot \left( \begin{bmatrix} 0 \\ 0 \\ . \\ . \\ . \\ 0 \\ w_1 \end{bmatrix} + \begin{bmatrix} 0 \\ 0 \\ . \\ . \\ . \\ 0 \\ w_2 \end{bmatrix} \right) \right) + \begin{bmatrix} 0 \\ 0 \\ . \\ . \\ . \\ 0 \\ w_2 \end{bmatrix} \right) ... \right)$$

$$= \sigma \left( ...\sigma \left( W \cdot \left( \begin{bmatrix} 0 \\ 0 \\ . \\ . \\ . \\ w_1 \\ w_2 \end{bmatrix} + \begin{bmatrix} 0 \\ 0 \\ . \\ . \\ . \\ 0 \\ w_3 \end{bmatrix} \right) \right) ... \right)$$

$$= ...$$

$$= \begin{bmatrix} w_1 \\ w_2 \\ . \\ . \\ . \\ w_{d-1} \\ w_d \end{bmatrix} = w$$

which is the input to the first of $k$ feed forward layers in $R_M$.
Since, the $k$ feed forward layers of $R_M$ are the same as in $M$, we have:

$$R_M(w) = M(w) \qquad \forall w \in \Sigma^d$$

Since answering counterfactual/MCR queries with regard to Hamming distance is NP complete for MLPs Barceló et al. (2020), the complexity of answering counterfactual queries for $R_M$ with regard to Hamming distance must also be NP hard. Thus, the complexity of evaluating counterfactual queries for RNN classifiers in general, with regard to Hamming distance is, NP hard.

For edit distance, we construct an RNN binary classifier $R'_M :=$ $\langle \Sigma', N, h_{-1}, W, W', \sigma, W''_{R'_M}, b_{R'_M}, f_{R'_M}, k_{R'_M} \rangle$ where,

- $\Sigma' := \{0, 1, \$_1, \$_2, ..., \$_d\}$,
- $N := \{1, ..., d, d+1, ..., 2d\}$,
- $h_{-1} \in \mathbb{Q}^{2d}$ is the zero vector,
- $W \in \mathbb{Q}^{2d \times 2d}$ is the upper shift matrix: $\begin{bmatrix} 0 & 1 & 0 & ... & 0 \\ 0 & 0 & 1 & ... & 0 \\ & & \vdots & & \\ 0 & 0 & 0 & ... & 1 \\ 0 & 0 & 0 & ... & 0 \end{bmatrix}$,
- $W'_0 := \begin{bmatrix} 0 & 0 & ... & 0 & -4 \end{bmatrix}^T \in \mathbb{Q}^{2d}$,
- $W'_1 := \begin{bmatrix} 0 & 0 & ... & 0 & -2 \end{bmatrix}^T \in \mathbb{Q}^{2d}$,
- $W'_{\$_j} := \begin{bmatrix} 0 & 0 & ... & 0 & j \end{bmatrix}^T \in \mathbb{Q}^{2d} \qquad \forall j \in \{1, ..., d\}$,
- $\sigma$ is the identity function,

- $W''^{(1)}_{R'_M} := \begin{bmatrix} 0 & 1 & 0 & 0 & ... & 0 & 0 \\ 0 & 0 & 0 & 1 & ... & 0 & 0 \\ & & & \vdots & & & \\ 0 & 0 & 0 & 0 & ... & 0 & 1 \\ \hline 1 & 0 & 0 & 0 & ... & 0 & 0 \\ 0 & 0 & 1 & 0 & ... & 0 & 0 \\ & & & \vdots & & & \\ 0 & 0 & 0 & 0 & ... & 1 & 0 \end{bmatrix} \in \mathbb{Q}^{2d \times 2d},$

- $W''^{(2)}_{R'_M} := \left[ \begin{array}{c|c} 0.5I & \mathbf{0} \\ \hline \mathbf{0} & I \end{array} \right] \in \mathbb{Q}^{2d \times 2d}$ and $I, \mathbf{0} \in \mathbb{Q}^{d \times d}$,

- $W''^{(3)}_{R'_M} := \left[ \begin{array}{c|c} I & \mathbf{0} \\ \hline \mathbf{0} & H \end{array} \right] \in \mathbb{Q}^{2d \times 2d}$ where,

  $H := \begin{bmatrix} 1 & -1 & 0 & 0 & ... & 0 & 0 \\ 0 & 1 & -1 & 0 & ... & 0 & 0 \\ & & & \vdots & & & \\ 0 & 0 & 0 & 0 & ... & 1 & -1 \\ 0 & 0 & 0 & 0 & ... & 0 & 1 \end{bmatrix} \in \mathbb{Q}^{d \times d}$, and $I, \mathbf{0} \in \mathbb{Q}^{d \times d}$

  ($H$ is the Identity Matrix minus the Upper Shift Matrix),

- $W''^{(4)}_{R'_M} := \left[ \begin{array}{c|c} I & \mathbf{0} \\ \hline 0 & 1 \end{array} \right] \in \mathbb{Q}^{(d+1) \times 2d}$, where $I, \mathbf{0} \in \mathbb{Q}^{d \times d}$, $0 \in \mathbb{Q}^{1 \times d}$ is the zero row vector and $1 \in \mathbb{Q}^{1 \times d}$ is the ones row vector

- $W''^{(p)}_{R'_M} := W''^{(p-4)} \quad \forall p \in \{5, ..., k+3\}$ (from $R_M$)

- $W''^{(k+4)}_{R'_M} := \left[ \begin{array}{c|c} W''^{(k)} & \mathbf{0} \\ \hline 0 & -1 \end{array} \right]$, where $0$ is the zero row vector, $\mathbf{0}$ is the zero column vector, $-1 \in \mathbb{Q}$ (and $W''^{(k)}$ is from $R_M$),

- $W''^{(k+5)}_{R'_M} := [0.5 \quad 0.5] \in \mathbb{Q}^{1 \times 2}$,

- $b^{(1)}_{R'_M} \in \mathbb{Q}^{2d}$ is the zero vector,

- $b^{(2)}_{R'_M} := \begin{bmatrix} \mathbf{2} & | & \mathbf{0} \end{bmatrix}^T \in \mathbb{Q}^{2d}$ (the first $d$ elements are 2 and the last $d$ elements are 0)

- $b^{(3)}_{R'_M} := \begin{bmatrix} \mathbf{0} & | & 1 & 1 & ... & 1 & -d \end{bmatrix}^T \in \mathbb{Q}^{2d}$ and $\mathbf{0} \in \mathbb{Q}^{1 \times d}$,

- $b^{(4)}_{R'_M} \in \mathbb{Q}^{(d+1)}$ is the zero vector,

- $b^{(p)}_{R'_M} := b^{(p-4)} \quad \forall p \in \{5, ..., k+3\}$ (from $R_M$),

- $b^{(k+4)}_{R'_M} := \begin{bmatrix} b^{(k)} & 0 \end{bmatrix}^T$ ($b^{(k)}$ is from $R_M$),

- $b^{(k+5)}_{R'_M} \in \mathbb{Q}$ is the zero vector,

- $f^{(1)}_{R'_M}, f^{(2)}_{R'_M} : \mathbb{Q}^{2d} \to \mathbb{Q}^{2d}$ are the identity function,

- $f^{(3)}_{R'_M} := f_{abs} : \mathbb{Q}^{2d} \to \mathbb{Q}^{2d}$ (the elementwise absolute value function) where,

  $f_{abs}(x)_i = \begin{cases} x_i & \text{if } x_i \geq 0 \\ -x_i & \text{otherwise} \end{cases},$

- $f^{(4)}_{R'_M}(x)_i := \begin{cases} 0 & \text{if } x_i = 0 \\ 1 & \text{otherwise} \end{cases}$, for $i \in \{1, .., d+1\}$, with $f^{(4)}_{R'_M} : \mathbb{Q}^{(d+1)} \to \mathbb{Q}^{(d+1)}$

- $f^{(p)}_{R'_M} := f^{(p-4)} \quad \forall p \in \{5, ..., k+3\}$,

- $f_{R'_M}^{(k+4)} := step$ is the elementwise applied step function, where
$$step(x)_i := \begin{cases} 0 & \text{if } x_i < 0 \\ 1 & \text{otherwise} \end{cases},$$

- $f_{R'_M}^{(k+5)} := f_q : \mathbb{Q} \to \mathbb{Q}$ where, $f_q(x) := \begin{cases} 1 & \text{if x = 1} \\ 0 & \text{otherwise} \end{cases}$

- $k_{R'_M} := k + 5$ (with $k$ from $R_M$)

For an arbitrary input $w := w_1, w_2, ..., w_d$ to $M$, we consider $w' := \$_1, w_1, \$_2, w_2, ...\$_d, w_d$ as input to $R'_M$. $R'_M$ is constructed such that it accepts an input $x$, iff it has the form $\$_1, x_1, \$_2, x_2, ...\$_d, x_d$ for some $x_1, ..., x_d$ and $x$ without delimiter symbols would be accepted by MLP $M$. For $R'_M$ the single layer recurrent network layer part is the same as that of $R_M$ while, the feed forward network part has $k + 5$ layers. Let's consider the behaviour of $R'_M$ on an input $w' := \$_1, w_1, \$_2, w_2, ...\$_d, w_d$. The single layer recurrent network of $R'_M$ converts the input sequence $w'$ into a vector $\begin{bmatrix} \$_1 & w_1 & \$_2 & w_2 & ... & \$_d & w_d \end{bmatrix}^T$. The vector is then converted into $\begin{bmatrix} w_1 & w_2 & ...w_d & \$_1 & \$_2 & ... & \$_d \end{bmatrix}^T$ using the first two layers of $k + 5$ feedforward layers. The next 2 layers check if the last $d$ elements of the vector are the ordered sequence of numbers from 1 to $d$. The next $k - 1$ layers process the first $d$ elements of the vector the same way as the MLP $M$ would have, without affecting the last element. The penultimate layer i.e. the $(k + 4)^{th}$ layer, processes all but the last element the same way as the last layer of MLP $M$ would have , and the last 1 element to produce 1 iff the $d$ delimiter symbols are in proper order, and 0 otherwise. The last layer is the $(k + 5)^{th}$ layer which produces the final output, it produces 1 only if the delimiters are in the proper order and $M$ would have accepted the input without delimiters.

If $v'$ and $w'$ are of this form where every other symbol is a delimiter and the delimiters are in proper order, and they have the same length, then $d_E(v', w') = d_H(v', w')$. This is because no sequence of delete, insert and substitute instructions, that ends with the same delimiter structure as it began, can incur a lower cost than just substituting the non-delimiter elements.

Let us now consider the counterfactual query. Given an arbitrary MLP $M$ with input from $\{0, 1\}^d$, we construct the RNN classifier $R'_M$ as above. Consider an arbitrary $w = w_1, ..., w_d \in \{0, 1\}^d$. Without limiting generality, we assume $w$ is not accepted by $M$. (Otherwise, we could add a layer to $M$ that flips the result before constructing $R'_M$.) Let $w' = \$_1, w_1, ..., \$_d, w_d$. Assume we have a polynomial time algorithm that answers the counterfactual query for $R'_M$ with input $w'$ with respect to edit distance. Assume this algorithm returns $v'$. We then know that $v'$ is accepted by $R'_M$, meaning $v' = \$_1, v_1, \$_2, v_2, ..., \$_d, v_d$ for some $v_1, ..., v_d \in \Sigma'$ and $v := v_1, .., v_d$ is accepted by $M$. We can further assume that $v_1, ..., v_d \in \{0, 1\}$ (for $v_i \notin \{0, 1\}$ we can set $v_i$ to 1, after computing $f_{R'_M}^{(4)}$ the result will be the same). Because $v'$ and $w'$ have the same length and both have the proper delimiters in the right order, we know that $d_E(v', w') = d_H(v', w')$. Furthermore, because the delimiter symbols do not need to be substituted, we have $d_H(v', w') = d_H(v, w)$. This means that $v$ is a valid answer to the counterfactual query of $M$ with input $w$ with regard to Hamming distance. Since that problem is NP complete for MLPs Barceló et al. (2020), the complexity of answering counterfactual queries for RNN classifiers in general, with regard to edit distance, is also NP hard.

$\square$

## D  TRANSFORMERS

The following definitions follow "Attention is Turing Complete" Pérez et al. (2021). A seq-to-seq network is defined as a function $N$, such that the value $N(X, s, r)$ corresponds to an output sequence of the form $Y = (y_1, y_2, ..., y_r)$. Furthermore:

**Definition D.1.** A *seq-to-seq language recognizer* is a tuple $A = (\Sigma, f, N, s, \mathbb{F})$, where $\Sigma$ is a finite alphabet, $f : \Sigma \to \mathbb{Q}^d$ is an *embedding function*, $N$ is a seq-to-seq network, $s \in \mathbb{Q}^d$ is a *seed vector*, and $\mathbb{F} \subseteq \mathbb{Q}^d$ is a set of *final vectors*. We say that $A$ *accepts the string* $w \in \Sigma^*$, if there exists an integer $r \in \mathbb{N}$ such that $N(f(w), s, r) = (y_1, ..., y_r)$ and $y_r \in \mathbb{F}$. The language accepted by $A$, denoted by $L(A)$, is the set of all strings accepted by $A$.

This definition comes with additional constraints:

- The embedding function $f : \Sigma \to \mathbb{Q}^d$ should be computed by a Turing machine in polynomial time w.r.t. the size of $\Sigma$.
- The set $\mathbb{F}$ should also be recognizable in polynomial-time; given a vector $\boldsymbol{f}$, the membership $\boldsymbol{f} \in F$ should be decided by a Turing machine working in polynomial time with respect to the size (in bits) of $\boldsymbol{f}$.

We then define a transformer:

**Definition D.2.** A *Transformer* $T$ is a tuple
$T = \langle score, \rho, L^{enc}, L^{dec}, \Theta^{enc}, \Theta^{dec}, K_{final}^{enc}, V_{final}^{enc}, F \rangle$, where

- $score : \mathbb{Q}^d \times \mathbb{Q}^d \to \mathbb{Q}$ is a *scoring function* with $d > 0$

- $\rho : \mathbb{Q}^n \to \mathbb{Q}^n$ is a *normalization function* with $n > 0$

- $L^{enc} \in \mathbb{N}$ is the number of encoding layers.

- $L^{dec} \in \mathbb{N}$ is the number of decoding layers.

- $\Theta^{enc} = \theta_1^{enc}, ..., \theta_{L^{enc}}^{enc}$ is a sequence of parameters, with $\theta_i^{enc} = (Q_i^{enc}, K_i^{enc}, V_i^{enc}, O_i^{enc})$, where $Q_i^{enc}, K_i^{enc}, V_i^{enc}, O_i^{enc}$ are functions from $\mathbb{Q}^d$ to $\mathbb{Q}^d$, for $i \in \{1, ..., L^{enc}\}$

- $\Theta^{dec} = \theta_1^{dec}, ..., \theta_{L^{dec}}^{dec}$ is a sequence of parameters, with $\theta_i^{dec} = (Q_i^{dec}, K_i^{dec}, V_i^{dec}, O_i^{dec})$, where $Q_i^{dec}, K_i^{dec}, V_i^{dec}, O_i^{dec}$ are functions from $\mathbb{Q}^d$ to $\mathbb{Q}^d$, for $i \in \{1, ..., L^{dec}\}$

- $K_{final}^{enc}, V_{final}^{enc}$ are final transformation functions of the encoder,

- $F$ is a final transformation function.

In order to define the output of a transformer, we first make some definitions for the different parts of a transformer.
Attention is defined with:

$$Att(\mathbf{q}, \mathbf{K}, \mathbf{V}) := s_1 \boldsymbol{v}_1 + s_2 \boldsymbol{v}_2 + \cdots + s_n \boldsymbol{v}_n, , \text{ where}$$
$$\mathbf{V} = \boldsymbol{v}_1, ..., \boldsymbol{v}_n \text{ and}$$
$$(s_1, \ldots, s_n) := \rho(score(\boldsymbol{q}, \boldsymbol{k}_1), score(\boldsymbol{q}, \boldsymbol{k}_2), \ldots, score(\boldsymbol{q}, \boldsymbol{k}_n))$$

The transformer consists of an encoder and a decoder, layer $l$ of the encoder is a function $Enc(\mathbf{X}, \theta_j^{enc})$, that maps a sequence of vectors in $\mathbb{Q}^d$, $\mathbf{X} = \boldsymbol{x}_1, ..., \boldsymbol{x}_n$ to a sequence of vectors in $\mathbb{Q}^d$, $\mathbf{Z} = \boldsymbol{z}_1, ..., \boldsymbol{z}_n$ with

$$\boldsymbol{z}_i = O_l^{enc}(\boldsymbol{a}_i) + \boldsymbol{a}_i$$

where $\boldsymbol{a}_i = Att(Q_l^{enc}(\boldsymbol{x}_i), K_l^{enc}(\mathbf{X}), V_l^{enc}(\mathbf{X})) + \boldsymbol{x}_i$ for $i \in \{1, ..., n\}$.
The $L^{enc}$-layer Transformer encoder is defined by the following recursion (with $1 \le l \le L^{enc} - 1$ and $\mathbf{X}^1 = \mathbf{X}$ the input of the encoder).

$$\mathbf{X}^{l+1} := Enc(\mathbf{X}^l; \boldsymbol{\theta}_l)$$

The result of the $L^{enc}$ layer encoder is: $TEnc_L(\mathbf{X}) := (K_{final}^{enc}(\mathbf{X}^{L^{enc}}), V_{final}^{enc}(\mathbf{X}^{L^{enc}}))$.
For layer $l$ we consider input $\mathbf{Y} = (\boldsymbol{y}_1, ..., \boldsymbol{y}_k)$, an external pair of key-value vectors $(\mathbf{K}^e, \mathbf{V}^e)$ and output $Dec((\mathbf{K}^e, \mathbf{V}^e), \mathbf{Y}; \theta_l^{dec}) := \mathbf{Z} = (\boldsymbol{z}_1, ..., \boldsymbol{z}_k)$, where for $i \in \{1, ..., k\}$:

$$\boldsymbol{p}_i = Att(Q_l^{dec}(\boldsymbol{y}_i), K_l^{dec}(\mathbf{Y}_i), V_l^{dec}(\mathbf{Y}_i)) + \boldsymbol{y}_i$$
$$\boldsymbol{a}_i = Att(\boldsymbol{p}_i, \mathbf{K}^e, \mathbf{V}^e) + \boldsymbol{p}_i$$
$$\boldsymbol{z}_i = O_l^{dec}(\boldsymbol{a}_i) + \boldsymbol{a}_i$$

Similar to the encoder we then define the decoder with:

$$\mathbf{Y}^{l+1} = Dec((\mathbf{K}^e, \mathbf{V}^e), \mathbf{Y}^l; \theta_l)$$

for $1 \le l \le L^{dec} - 1$ and $\mathbf{Y}^1 = \mathbf{Y}$ the input.
The result of the entire decoder is: $TDec_L(\mathbf{Y}) := F(\boldsymbol{y}_k^{L^{dec}})$

For an input $\boldsymbol{X}$, a seed vector $\boldsymbol{y}_0$, and a value $r \in \mathbb{N}$, we define the result of the entire transformer $Trans(\boldsymbol{X}, \boldsymbol{y}_0, r) := \boldsymbol{Y} = (\boldsymbol{y}_1, ..., \boldsymbol{y}_r)$ with

$$\boldsymbol{y}_{t+1} = TDec_L(TEnc_L(\boldsymbol{X}), (\boldsymbol{y}_0, \boldsymbol{y}_1, \ldots, \boldsymbol{y}_t)), \qquad \text{for } 0 \le t \le r-1.$$

Let $pos : \mathbb{N} \to \mathbb{Q}^d$ be a positional encoding function. Before an input of position $i$ is processed by the encoder or decoder $pos(i)$ is added.

### D.1 PROOF OF PROPOSITION 1 FOR TRANSFORMER

*Proof. Hamming(1)/edit distance(2):*
In NP: For a given input $v \in \Sigma^*$, transformer $A$ as a seq-to-seq language recognizer, and $k \in \mathbb{N}$, guess $w$ such that $A(w) \neq A(v)$ and the distance between $d(v, w) \le k$.
Because Hamming distance can be computed in polynomial time and classification by $A$ can be computed in polynomial time, we can verify an answer $w$ to the counterfactual query with regards to Hamming distance in polynomial time.
For edit distance we do the same, except that we also guess a sequence of at most $k$ instructions (substitute, insert or delete), that when applied to $v$ yield $w$. Using these instructions, we can easily verify in polynomial time if $d(v, w) \le k$.

NP hard: Let $M$ be an arbitrary MLP $M$, that maps a vector from $\{0,1\}^D$ to $\{0,1\}$. We construct a seq-to-seq language recognizer (as defined by Pérez et al. (2021)) $A = (\Sigma, f, N, s, \mathbb{F})$, with $\Sigma := \{0,1\}^D$, $f : \Sigma \to \mathbb{Q}^1$ where $f(c) = M(c) \; \forall c \in \Sigma$, $N$ is any transformer with $N(0) \neq N(1)$, and $\mathbb{F} := \{N(1)\}$. By definition, we have: $A(w) = M(w), \forall w \in \{0,1\}^D$ Since answering counterfactual queries / the MCR problem with regards to Hamming distance is NP complete for MLPs Barceló et al. (2020), answering counterfactual queries for $A$ with regard to Hamming distance must also be NP hard. Thus, answering counterfactual queries for Transformers as seq-to-seq language recognizers in general, with regard to Hamming distance is NP hard.

This proof relies on the encoding function $f$ doing the computation of $M$. While this is in line with the definition, we would like to note that the Transformer itself (usually) also includes feed forward layers and can therefore do computations of at least the same complexity as an MLP.

*DTW*(3): Given an arbitrary Turing Machine $M$, we assume, without loss of generality, that it does not halt after 0 or 1 steps. Let $L_M \subseteq \{a\}^*$ be empty if $M$ does not halt on the empty input, and $\{a^t\}$ otherwise, where $t$ is the number of steps, $M$ took to halt. Asking "Is $L_M \neq \emptyset$ ?" is equivalent to asking "Does $M$ halt?", which is semi-decidable. Therefore, by the proof of Theorem 6 in "Attention is Turing Complete" (Pérez et al., 2021), there exists a Transformer $T$, that recognizes $L_M$. Since that proof is constructive, we can obtain $T$. We know that for $\Sigma = \{a\}$, we have $DTW(w, v) = 0$ for all $w, v \in \Sigma^+$. Let $w := a$, a non-empty word over $\{a\}$, with $w \notin L_M$. Answering the counterfactual query for $w$ on $T$, is equivalent to checking emptiness of $L_M$. If counterfactual queries are decidable, then so is non-emptiness of $L_M$, which implies decidability of the halting problem. Since the Halting Problem is known to be undecidable, checking the existence of a counterfactual explanation for transformers must also be undecidable. $\qquad\square$

## E MISSING PARTS OF PROOF OF THEOREM 1

**Hamming distance** The required changes are the construction of the weighted graph $G = (V, E, S, T)$. The sets $V$, $S$, and $T$ of vertices in the proof remain the same. What changes is the edge relation $E$. In particular, for each transition $(p, P, q) \in \Delta$ with $P = [r, s]$, we add an edge from $(p, i)$ to $(q, i+1)$, whose weight is 0 if $a \in P$, otherwise its weight is 1. The rest of the proof remains the same.

**Edit distance** Like for Hamming distance, the required changes here are also just the construction of the edge relation $E$. We take a distance $d$ for elements; the proof works, either when $d$ just checks for equality for elements (i.e. $d(a, b) = 0$ if $a = b$; otherwise, $d(a, b) = \infty$) or for the distance measure $|a - b|$. For each transition $(p, P, q) \in \Delta$ with $P = [r, s]$, we add the following edges to $E$:

1. *matching*: $(p, i)$ to $(q, i+1)$ with weight $\inf_{t \in P} d(t, a_i)$. This can be computed using linear programming just like for the proof for the case of DTW.

2. *Skip $v$'s current element*: $(p, i)$ to $(p, i + 1)$, whose weight is $c_{\text{insdel}}$.

3. *Skip $w$'s current element*: $(p, i)$ to $(q, i)$, whose weight is $c_{\text{insdel}}$.

The rest of the proof remains the same.

**Dealing with open intervals**   We sketch this only for DTW and the proof is the same for the rest. In the case when not all intervals are of the form $[s, t]$, our set of vertices is of the form $V = X \times \{c, o\}$, where $X = Q \times \{1, \dots, n\}$. Here, $o$ indicates that the closed part of a non-closed interval has been "used". The initial set $S$ of vertices is $\{q_0\} \times \{1\} \times \{c\}$, indicating that no open intervals have been used. We will make two queries with the first target set $T$ to be $F \times \{n\} \times \{c\}$ and the second target set $T'$ to be $F \times \{n\} \times \{o\}$. We will then compare the value of the minimal weight $W$ for $T$ and $T'$. Three cases are under consideration, as follows. If $W$ is met by $T$, then we compare $W \leq k$. If $W$ is met by $T$, then we compare $W < k$. If $W$ is met by both $T$ and $T'$, then we compare $W \leq k$.

For each $(p, P, q) \in \Delta$, we add the following edges:

1. $(p, i, c)$ to $(q, i, c)$ with weight $W := \inf_{r \in P} |r - a_i|$ when $W \in P$.

2. $(p, i, ?)$ to $(q, i, o)$ with weight $W := \inf_{r \in P} |r - a_i|$ when $W \notin P$ and $? \in \{c, o\}$.

3. $(p, i, c)$ to $(q, j, c)$ for every $j \in \{i + 1, \dots, n\}$ with weight $W := \inf_{t \in P} \sum_{h=i}^{j-1} |t - a_h|$, when $W \in P$.

4. $(p, i, ?)$ to $(q, j, c)$ for every $j \in \{i + 1, \dots, n\}$ with weight $W := \inf_{t \in P} \sum_{h=i}^{j-1} |t - a_h|$, when $W \notin P$ and $? \in \{c, o\}$.

In particular, for the second and fourth cases (i.e. when $W \notin P$), we impose that the final check is a strict inequality.

# F   HARDNESS PROOFS FOR NFA

**Proposition 4.** *Evaluating counterfactual queries for NFAs under Hamming distance or Edit distance is NP-hard.*

*Proof.* We reduce from CNF-Satisfiability in two steps: (1) We show coNP-hardness of linear-length universality of NFAs, specifically the following problem: given an $n$-state NFA $M$ over $\Sigma$, does $M$ accept all strings of length $n$, i.e., is $\Sigma^n \subseteq L(M)$? For related problems, we refer to Gawrychowski et al. (2020). (2) We then reduce linear-length universality of NFAs to evaluating counterfactual queries for NFAs under the Hamming distance and edit distance.

For (1), we reduce from CNF-Satisfiability as follows: Given an arbitrary CNF formula $\phi$ with $n$ variables and $m$ clauses, we create, using De Morgan's laws, a DNF formula $\phi'$ of the same size such that $\phi'$ is equivalent to the negation of $\phi$. We then construct an NFA $M_{\phi'}$ over $\{0, 1\}$ with $O(nm)$ states that accepts a string $w \in \{0, 1\}^n$ if and only if the variable setting $x_i = w_i, i \in [n]$ satisfies $\phi'$. To do so, we create states $q_{j,0}, \dots, q_{j,n}$ for all $j \in [m]$, where we unify the states $q_{j,0}, j \in [m]$ to become the starting state $q_0$. For each clause $C_j$ and variable $x_i$, we introduce transition(s) from $q_{j,i-1}$ to $q_{j,i}$ as follows: if $x_i$ is a literal in $C_j$, the transition is labelled with 1, if $\overline{x_i}$ is a literal in $C_j$ it is labelled with 0, and if $x_i$ and $\overline{x_i}$ do not occur as literals in $C_j$, the transition is labelled with both 0 and 1 (we can assume without loss of generality that $x_i$ and $\overline{x_i}$ do not occur both as literals). All states in this NFA are accepting. Note that the constructed NFA $M_{\phi'}$ accepts $w \in \{0, 1\}^n$ if and only if there exists some $j$ such that we can traverse the path from $q_{j,0}$ to $q_{j,n}$, which is equivalent to $C_j$ being satisfied by $w$. Thus, $M_{\phi'}$ accepts all strings in $\{0, 1\}^n$ if and only if $\phi'$ is satisfied by all $x \in \{0, 1\}^n$, which is equivalent to $\phi$ being unsatisfiable. By adding self-loops to all $q_{j,n}, j \in [m]$ with labels 0 and 1, we observe that the resulting NFA has $N = nm + 1$ states and accepts all strings of length $N = nm + 1 \geq n$ if and only if $\phi$ is unsatisfiable. Clearly, the reduction runs in polynomial time, yielding coNP-hardness of linear-length NFA universality.

For (2), the reduction from linear-length universality of NFAs to evaluating counterfactual queries for NFAs under the Hamming distance is immediate: Given an $n$-state NFA $M$ over $\Sigma$, we first compute whether $v = 0^n$ is rejected by $M$. If so, we reject the given instance. Otherwise, we evaluate the

counterfactual query for $M$, $v = 0^n$ and $k = n$. Note that there exists some $w$ of Hamming distance at most $n$ to $v$ that is a counterfactual to $x$, i.e., rejected by $M$, if and only if $\Sigma^n \nsubseteq L(M)$.

To adapt the argument to the edit distance, we will slightly adapt the given NFA $M$ to a new NFA $M'$: Let us denote the starting state of $M$ as $q_0$. We will add new states $q_1, \ldots, q_{2n}$ and, for each $1 \le i \le 2n$ and $\sigma \in \Sigma$, a transition from $q_{i-1}$ to $q_i$ labelled $\sigma$. By making each state $q_i$ *except* $q_n$ accepting, this ensures that $L(M') = L(M) \cup \bigcup_{0 \le i \le 2n, i \ne n} \Sigma^i$. We now proceed analogously to before: If $M$ rejects $xv0^n$, we reject the given linear-length NFA universality instance. Otherwise, we perform the counterfactual query for $M'$, $v = 0^n$ and $k = n$ over the edit distance. Note that there exists some $w$ with edit distance at most $n$ from $v$ that is rejected by $M'$ if and only if there exists some $w \in \bigcup_{i=0}^{2n} \Sigma^i$ that is rejected by $M'$. By construction of $M'$, the latter is equivalent to the existence of some $w \in \Sigma^n$ that is rejected by $M$. This yields a polynomial-time reduction, concluding the claim. $\square$

**Proposition 5.** *Evaluating counterfactual queries for NFAs under DTW distance is PSPACE-hard. It is even PSPACE-hard whenever there exists $\sigma, \sigma' \in \Sigma$ such that $\sigma \ne \sigma'$ and $c(\sigma, \sigma') = 0$.*

*Proof.* To show this, we define $\Sigma$ so that there exist $\sigma, \sigma' \in \Sigma$ such that $\sigma \ne \sigma'$ and $c(\sigma, \sigma') = 0$.

Classic results show that NFA universality over $\Sigma$ (given an NFA $M$ over $\Sigma$, determine whether $L(M) = \{0,1\}^*$) is PSPACE-complete already when $\Sigma = \{0, 1\}$ III et al. (1976) and coNP-complete if $\Sigma = \{0\}$ Stockmeyer & Meyer (1973). Thus, consider DTW with any distance function $c : \Sigma \times \Sigma \to \mathbb{N}$ such that there exists $\sigma, \sigma' \in \Sigma$ with $\sigma \ne \sigma'$ and $c(\sigma, \sigma') = 0$. We reduce from NFA universality over $\{0, 1\}$ as follows: Given an NFA $M$ over $\{0, 1\}$, we first check if $v = \sigma$ is rejected by $M$. If so, we reject. Otherwise, we perform the counterfactual query on an NFA $M'$, $v = \sigma$ and $k = 0$, where $M'$ is obtained from $M$ by replacing every 0-label of a transition by $\sigma$ and every 1-label by $\sigma'$, as well as adding a branch to the NFA that accepts every word containing at least one symbol different from $\sigma$ and $\sigma'$. Note that the set of strings of DTW distance 0 from $\sigma$ consists of (1) all strings in $\{\sigma, \sigma'\}^*$ as well as (2) possibly additional strings containing at least one symbol different from $\sigma$ and $\sigma'$. Since strings of category (2) are trivially accepted by $M'$, there exists a string $w$ that has DTW distance 0 from $v$ and is rejected by $M'$ if and only if $L(M') \nsupseteq \{\sigma, \sigma'\}^*$ which is equivalent to $L(M) \nsupseteq \{0,1\}^*$. The reduction is computable in polynomial time, concluding PSPACE-hardness of evaluating counterfactual queries for NFAs under DTW distance.

For the case that no distinct symbols $\sigma, \sigma'$ with $c(\sigma, \sigma') = 0$ exists, we still obtain NP-hardness under the natural assumption that there exists $\sigma \in \Sigma$ with $c(\sigma, \sigma) = 0$ (typically, any reasonable distance function for DTW satisfies $c(\sigma, \sigma) = 0$ for *all* $\sigma \in \Sigma$). We adapt the reduction from above to reduce from NFA universality over $\{0\}$: Given an NFA over $M$, we construct $M'$ analogously to above so that $M'$ accepts all strings containing at least one symbol different from $\sigma$, as well as any string $\sigma^i$ if and only $M$ accepts $0^i$. Since we may assume without loss of generality that $\sigma$ is rejected by $M$, the resulting counterfactual query $M'$, $v = \sigma$, $k = 0$ is empty if and only if $L(M) = \{0\}^*$, concluding the claim. $\square$

