# OpenReview forum: "Complexity of Formal Explainability for Sequential  Models"
_ICLR.cc/2024/Conference — Submitted to ICLR 2024_

### Official Review · Reviewer_XaKE · 2023-10-31

**Soundness:** 3 good
**Presentation:** 2 fair
**Contribution:** 2 fair
**Rating:** 5
**Confidence:** 4

**Summary:**

This purely theoretical paper is devoted to formal explainability of
sequential ML models, e.g. RNNs and Transformers. Namely, the paper
lifts the apparatus of formal XAI (FXAI) to sequential models and
studies the problem of checking whether there exists a single formal
abductive explanation or a single formal contrastive explanation for
such a model's prediction and offers a few theoretical results
demonstrating the complexity of this problem. Then the authors discuss
the complexity of the same problem for various types of automata
showing some negative (intractability) and some positive (polytime)
results in this domain.

**Strengths:**

- This paper discusses the complexity of testing whether there exists
  a formal explanation for a sequential ML model. Although this is not
  the first work on the subject (I've already reviewed a few this
  year), but the theoretical contributions seem clearly novel to me.

- The paper looks rigorous in terms of the problems studied and
  propositions claimed. I haven't checked the proofs but the results
  make sense to me.

**Weaknesses:**

- This is a purely theoretical paper with a list of propositions and
  proof sketches but with absolutely no practical side of things. No
  experimental evidence is provided either. As a result, whether or
  not FXAI is usable in the context of such models is left unclear.
  (Recall that AXp's and CXp's are known to be in general intractable
  to compute for non-sequential models; so no big deal.) This makes me
  wonder why this line of work can be deemed relevant for a conference
  like ICLR. (KR would be a nice fit but ICLR is hard to justify.)

- On a similar note: overall presentation could and should be
  significantly improved. Definitions of abductive and contrastive
  explanations in the context of sequential models are given in a
  somewhat lazy way. (The models are defined similarly.) Also, the
  paper gives no examples whatsoever that would demonstrate how these
  explanations look like for sequential models. As this paper claims
  to be the first of its kind, such examples should be provided.

- No algorithms to be used for computing such explanations are
  proposed (even in theory). This exacerbates the issue of the venue
  choice above.

- Minor: on p9 there is a sentence: "This type of explainability falls
  within the category of post-hoc local model-agnostic explanations".
  No, it does not. Both abductive and contrastive explanations are
  global. None of them are model-agnostic as you have to reason about
  the model's behavior formally and hence require access to its
  original representation.

**Questions:**

- Can you show a clear example of an RNN model, some of the instances
  it takes as input and also exemplify abductive and contrastive
  explanations for some of the model's decisions?

- Are the algorithms originally proposed for AXp and CXp computation
  are directly usable in this setting? If not, how should one compute
  them?

- Can you summarize the differences between the complexity of these
  problems for non-sequential and sequential models? This way, can you
  contrast your results with what is already known to hold in general?

- Why do you need 1 ** k in the informal definition of a CXp? It
  doesn't seem to be used anywhere, does it?

---

> ### Author Response · Authors · 2023-11-22
>
> Q: No algorithms to be used for computing such explanations are proposed (even in theory).
> A: Thank you for bringing up the question of computing explanations. We have stated our positive results as polynomial-time algorithms for decision problems. However, the algorithms can actually generate (counterfactual) explanations in polynomial-time. We apologize for not making this explicit, which we have corrected in the revised version.
>
> Q: How are they related to machine learning, FXAI, and ICLR?
> A: We thank you for your questions and comments. We would like to convince you that our results are of interests to those working in FXAI and ML. In the revised version, we have placed a stronger emphasis on the connection to model generation (learning) and explanation generation. Note that our results concern the latter, so we mention relevant existing results on the former to put the results in contexts. We summarize the connection to ML below.
>
> Our work was actually very much motivated by the machine learning work on the extraction of DFA from RNN using automata learning algorithms (e.g. Angluin's L*), e.g., see the ICML paper by Weiss et al. (2018) and the AAAI paper by Okudono et al. (2020). This work hinges on the assumption that DFA is *more interpretable* than RNN, but it was never really clear what this means. Following the work for non-sequential models, we associate interpretability with the computational tractability (following standard complexity theoretic assumptions) to compute abductive and counterfactual explanations. In this technical sense, we showed that DFA is more interpretable than RNN and transformers.
>
> So, a fuller working (F)XAI schema for could be as follows: (1) train RNN R, (2) extract DFA A out of R (e.g. using Weiss et al.), and (3) use our algorithm on A to generate explanations.
>
> Note that (1) could perhaps be replaced directly by an algorithm that extracts a DFA A directly from the datasets. Note also that although we deal with decision problems, the polynomial-time algorithms can be used to extract an explanation (we have clarified this in the new version).
>
> The aforementioned concentrates on a finite alphabet. Since time series (e.g. stock price chart) is ubiquitous in applications, we went one step further to look at sequential models that may take *data sequences* as input (i.e. sequences of rationals). Two prominent extensions of DFA in this setting are interval automata and k-deterministic register automata, for both of which automata learning algorithms (e.g. Moerman et al. (2017); Howar et al. (2012); Cassel et al. (2016); Drews & D’Antoni (2017); Argyros & D’Antoni (2018)) are available. In the new version, we have also expanded our small examples in the previous version on stock price signals, and how they can be captured by Interval Automata and 2-DRA. Finally, our results are a fundamental building block that enables a similar working schema to the aforementioned case of finite alphabet.
>
> Q: Are your explanations local and model agnostic?
>
> A: Local explanations refer to less complex solution subspaces (e.g. specific inputs), e.g., see survey https://arxiv.org/abs/1910.10045. We believe that both abductive and counterfactual explanations are local explanations since they refer to a specific input instance. The same claim was also made in previous works on such explanations (e.g. see the citation Barcelo et al. (2020) in our paper).
>
> Model-agnostic explanations treat the model as a blackbox, as far as the explanations. See the survey https://www.frontiersin.org/articles/10.3389/fdata.2021.688969/full. They explicitly pinpoint that counterfactual explanations are model-agnostic.
>
> Q: a clear example of a model with AXp and CXp explanations?
>
> A: In the new version, we have expanded our small examples in the previous version (Example 1 to Example 3). They concern a simple document identification task (using DFA over finite alphabets) and a simple stock trading signal (using interval automata/k-DRA). We show how counterfactual/abductive explanations mean in this context.
>
> Q: Are the algorithms originally proposed for AXp and CXp computation are directly usable in this setting? If not, how should one compute them?
>
> A: Existing algorithms for AXp and CXp concern non-sequential models, so cannot be used in our setting. Our polynomial-time algorithms can generate the counterfactuals in polynomial-time. We made this explicit in the revised version.
>
> Q: Differences between the complexity of these problems for non-sequential and sequential models?
>
> A: From existing works (e.g. by Izza et al. and Barcelo et al.), the complexity of explainability is typically decidable (within PSPACE) for non-sequential models. In contrast, the same problems are undecidable in the sequential setting, owing to Turing-completeness of RNN and Transformers.
>
> Q: 1 ** k in the informal definition of a CXp?
>
> A: This means the number is represented in unary. We clarified in this in the revised version.

---

> ### Comment · Reviewer_XaKE · 2023-11-22
> **Reply to the authors**
>
> #### **On relevance to FXAI and ICLR:**
>
> I did not say the paper was irrelevant to FXAI. What I say is that this purely theoretical contribution is completely detached from practice and hence is beyond the scope of ICLR. There are other conferences where this line of work would be a perfect fit. And I still believe that.
>
> #### **On newly added examples:**
>
> Thank you. Based on a brief look, abductive explanations seem to be defined to replicate the exact meaning of those for classification models, i.e. *as long as these features are included in the instances, the prediction is going to stay*. Unfortunately, there is no example of contrastive explanation but I presume it would look again similarly to the case of classification models.
>
> #### **On local and model-agnostic explanations:**
>
> Well, abductive explanations are global in the sense that they represent a prime implicant of the classification function despite being defined for a given instance. They are perfectly applicable to any other instance they are compatible with. The same hold for contrastive explanations. This assuming we are talking about formal XAI. As for model-agnostic, you have to formally reason about your ML model of interest and so you certainly need to have *white-box* access.
>
> #### **On inapplicability of existing algorithms:**
>
> Why can't they be used assuming you can represent your model logically and have a reasoner for it? Granted the complexity of the problem is different and this will affect how difficult the reasoning calls would be, the algorithms should still be applicable.
>
> #### **On everything else:**
>
> Thank you for the comments.

---

> > ### Author Response · Authors · 2023-11-22
> >
> > Thank you for the quick response.
> >
> > Q: Examples of contrastive explanations
> >
> > A: Our paper focuses on abductive and *counterfactual* explanations in the paper, *not* contrastive (which is by the way dual of abductive). Examples of counterfactual explanations are given in Example 1 and Example 2.
> >
> > Q: Applicability of existing algorithms on our problems
> >
> > A: We apologize that we are not very quick on understanding what you mean that existing algorithms can be used. Which existing algorithms do you mean and which logical representation? Also, are you referring to both abductive and counterfactual explanations? We would not be able to respond to this comment if we do not understand the context better. Many thanks in advance.
> >
> > Q: local/model-agnostic explanations
> >
> > A: thank you for your clarification. Could you please also say if counterfactual explanations (the main subject of our paper) are global and non model-agnostic?  Many thanks in advance.

---

### Official Review · Reviewer_ynBt · 2023-11-01

**Soundness:** 4 excellent
**Presentation:** 4 excellent
**Contribution:** 2 fair
**Rating:** 6
**Confidence:** 4

**Summary:**

This paper studies the computational complexity of explainability questions, both abductive and counterfactuals, over sequential models ranging from finite automata to Transformers.  All questions are computationally hard for RNNs and Transformers, but some are tractable for DFAs, and the paper shows that DFAs can be extended while keeping polynomial time complexity.

**Strengths:**

- The paper is well written and it is easy to read; the structure flows well and the body presents the right amount of detail about the proofs and technical aspects.
- The appropriate work seems properly referenced and presented.
- The paper presents novel results and focuses on sequential models, which are arguably one of the most important classes of models at the moment, and yet most formal XAI focuses on classification models.
- Proofs seem to be sound; I fully checked a couple of them and skimmed through the rest and they all seemed to be correct.

**Weaknesses:**

- In terms of presentation I think a small table summarizing all results would be helpful; the complexity landscape of the different queries is spread through the text right now.
- The main weakness of the paper in my opinion is that the complexity analysis doesn't seem to provide much new insight into the models or the explanations themselves. The counter to this point is that the presented extensions to DFAs seem like an interesting direction inspired by the complexity analysis. Nonetheless, there is no experimental evidence showing that they can be useful in practical prediction tasks, or directions towards making them more practical.

**Questions:**

I don't have particular questions besides the points mentioned as weaknesses. Perhaps something that could help the evaluation of the paper is some justification for this work being fit for the venue; at least superficially it seems of a slightly different nature than most ICLR work.

---

> ### Author Response · Authors · 2023-11-22
>
> Q: Perhaps something that could help the evaluation of the paper is some justification for this work being fit for the venue; at least superficially it seems of a slightly different nature than most ICLR work.
>
> A:  We thank you for your comment and question. In the new version, we have placed a stronger emphasis on the connection to model generation (learning) and explanation generation. Note that our results concern the latter, so we mention relevant existing results on the former to put the results in contexts. We summarize the connection to ICLR work below.
>
> Our work was actually very much motivated by the machine learning work on the extraction of DFA from RNN using automata learning algorithms (e.g. Angluin's L*), e.g., see the ICML paper by Weiss et al. (2018) and the AAAI paper by Okudono et al. (2020). This work hinges on the assumption that DFA is "more interpretable" than RNN, but it was never really clear what this means. Following the work for non-sequential models, we associate interpretability with the computational tractability (following standard complexity theoretic assumptions) to compute abductive and counterfactual explanations. In this technical sense, we showed that DFA is more interpretable than RNN and transformers.
>
> So, a fuller working (F)XAI schema for could be as follows: (1) train RNN R, (2) extract DFA A out of R (e.g. using Weiss et al.), and (3) use our algorithm on A to generate explanations.
>
> Note that (1) could perhaps be replaced directly by an algorithm that extracts a DFA A directly from the datasets. Note also that although we deal with decision problems, the polynomial-time algorithms can be used to extract an explanation (we have clarified this in the new version).
>
> The aforementioned work so far concentrates over strings (over a finite alphabet). Since time series (e.g. stock price chart) is ubiquitous in multiple applications, we went one step further to look at sequential models that may take "data sequences" as input (i.e. sequences of rationals). Two prominent extensions of DFA in this setting are interval automata and k-deterministic register automata, for both of which automata learning algorithms (e.g. Moerman et al. (2017); Howar et al. (2012); Cassel et al. (2016); Drews & D’Antoni (2017); Argyros & D’Antoni (2018)) have been developed in the last 10 years. In the new version, we have also expanded our small examples in the previous version on stock price signals, and how they can be captured by Interval Automata and 2-DRA. Finally, our results are a fundamental building block that enables a similar working schema to the aforementioned case of finite alphabet.
>
> In the revised version, we have also expanded our small examples in the previous version (Example 1 to Example 3). They concern a simple document identification task (using DFA over finite alphabets) and a simple stock trading signal (using interval automata/k-DRA). We show how counterfactual/abductive explanations mean in this context.
>
> Q: The main weakness of the paper in my opinion is that the complexity analysis doesn't seem to provide much new insight into the models or the explanations themselves. The counter to this point is that the presented extensions to DFAs seem like an interesting direction inspired by the complexity analysis. Nonetheless, there is no experimental evidence showing that they can be useful in practical prediction tasks, or directions towards making them more practical.
>
> A: We would like to convince you that our results can in fact provide insights into the models. We have stated our positive results as polynomial-time algorithms for decision problems. However, the algorithms can actually generate (counterfactual) explanations in polynomial-time. We have made this more explicit in the revised version.
>
> Although the focus of our current paper is on classifying explainability of different non-sequential models and, we do agree that providing experimental evidence of polynomial-time algorithms in the paper is an important next step. Another interesting research direction is how to realize the aforementioned (F)XAI schema for sequential data, by leveraging existing automata learning algorithms for DFA, interval automata, and k-DRA.
>
>
> Q: a small table summarizing the result would be useful
>
> A: Thanks for the excellent suggestion. We will add this in the final version.

---

> > ### Comment · Reviewer_ynBt · 2023-12-02
> >
> > After reading the discussions with the other reviewers, I have decided to update my score to a 6. I still believe the paper is a contribution worthy of publication, but I'm no longer convinced it meets the bar of significance for ICLR.  Constructively, I think a theoretical paper like this could/should have made a more compelling case for its significance tailored to the venue, as in its current state it seems to me like it is a better fit for other venues (e.g., KR, IJCAI, AAAI).

---

### Official Review · Reviewer_rsdT · 2023-11-02

**Soundness:** 3 good
**Presentation:** 3 good
**Contribution:** 2 fair
**Rating:** 6
**Confidence:** 3

**Summary:**

This paper considers the complexity of abductive and counterfactual explanations in sequence models such as RNNs, Transformers, DFAs, interval automata, and register automata. Here an abductive explanation is a form of minimal property test that shows when the output of the sequence model would change. Similarly, a counterfactual example is a "distance" bounded input sequence that results in a different output.

The results for RNNs and Transformers are mostly negative, following well known issues with Turing completeness. However, motivated by work on extracting automata from neural models, the paper studies the complexity of such explanations in the aforementioned automata classes. They provide a series of theorems and proofs which characterize the complexity.

The key results are the PSPACE completeness of abductive explanations for DFAs and the efficiency of counterfactual explanations for the studied automata classes (DFAs, NFAs, interval, and register).

**Strengths:**

1. The paper does a good job laying out the theoretical machinery needed to understand the abductive and counterfactual explanations. Further, when present (see below) the technical material seems to be sound, with the proofs being easy to follow.

2. The topic of explainability of sequence models is certainly important.

3. The classes of explanations discussed have utility and are motivated by similar definitions in the corresponding literature.

**Weaknesses:**

1. The exposition of the paper would greatly benefit from worked examples. It took a bit of studying to fully internalize the definitions presented in the paper and that is given a fair amount of automata theoretic background.

2. Unless I'm missing something, the proof of Prop 3. seems incomplete! The sentence ends with "This is equivalent to checking". I imagine this was an oversight during submission.

3. I'm actually having a hard time reconciling Prop 3. with Prop 2. In particular,  the paper rightfully points out that interval automata strictly generalize DFAs. Then shouldn't the results of PSPACE-hardness transfer? Otherwise, I could life the properties F and target DFA to the corresponding singleton interval automata, solve. This seems like a polynomial reduction?

4. Regarding the proof Prop 2, isn't DFA intersection / minimization poly time -- intersection is at product construction and minimization is polytime due to hopcroft's algorithm. So testing emptiness should be polytime. Why does the proof imply PSPACE-completeness?

*Note* If these are misunderstandings or easily corrected, I'm happy to revise my score.


# ----------------------- Update -----------------------

The authors have corrected some small technical issues with proof / latex and reconciled the soundness issues. Further examples have been added to the exposition.

I am raising my recommendation to weak accept and the corresponding soundness / presentation subscores.

The primary remaining issue is whether this is the right venue for this submission. Having been on the other side of such comments from reviewers, I acknowledge the subjective nature and am willing to cede after discussion with other reviewers.

**Questions:**

Additional questions:

1. See points 3 & 4 in weaknesses.
2. Do the negative results for non-determinism resolve if restricting to residual automata? They have many of the properties of DFAs while still admitting non-determinism.

---

> ### Author Response · Authors · 2023-11-22
>
> Q: The exposition of the paper would greatly benefit from worked examples. It took a bit of studying to fully internalize the definitions presented in the paper and that is given a fair amount of automata theoretic background.
>
> A: In the revised version, we have expanded our small examples of concepts (e.g. stock price signal) that can be captured using various notions of automata, and intuitions on what abductive/counterfactual explanations mean for such examples. In the final version, we would be happy to also provide a worked example of the construction.
>
> Q: Proposition 3 is confusing
> A: We thank you for pointing this out, and sincerely apologize for the confusion. There appeared to be a problem with a LaTeX macro that caused this part (Proposition 3 and incomplete proof) to be displayed when we switched into a submission mode. Certainly, this was not meant to be there, which we have deleted in the revised version.
>
> Q: Regarding the proof Prop 2, isn't DFA intersection / minimization poly time -- intersection is at product construction and minimization is polytime due to hopcroft's algorithm. So testing emptiness should be polytime. Why does the proof imply PSPACE-completeness?
>
> A: Indeed, checking emptiness of the intersection of two DFA is solvable in polynomial-time, but this problem becomes PSPACE-complete if we want to check emptiness of n DFAs (where n is non-fixed number). This was proven by Kozen in the FOCS'77 paper titled "Lower bounds for natural proof systems."
>
> Q: What happens with residual automata?
>
> A: From the point of view of learning, RFSA certainly helps because there is an extension of Angluin's L* learning algorithm for it (by Denis et al. (2001), and there is a recent result in the case of register automata in the paper titled "Residual Nominal Automata" by Moerman and Sammartino (2020)). However, in terms of explainability, we need to be able to complement the learned automaton in polynomial-time, and we do not know if RFSA can be complemented in polynomial-time (i.e. unlike DFA). So, it is not clear if counterfactual queries can be explained in polynomial-time as well for RFSA, which we believe is an interesting future research direction (similarly also, restrictions of NFA like unambiguous NFA), given the existence of an automata learner.

---

### Official Review · Reviewer_hXKh · 2023-11-03

**Soundness:** 4 excellent
**Presentation:** 4 excellent
**Contribution:** 2 fair
**Rating:** 6
**Confidence:** 4

**Summary:**

This paper studies the theoretical problem of generating abductive explanations and counterfactual explanations of ML models. The paper establishes negative results for extracting these explanations from RNNs and Transformers and then focuses on automata models, for which some PSPACE results can be extracted.

Overall, this seems like a solid theory contribution, but I don't see the relevance to ICLR.

**Strengths:**

Solid, clean automata theory.

**Weaknesses:**

1. The class of explanations here fail to convince me that any of the XAI goals are achieved.
2. Even if explanations could be generated for the purpose of checking them with formal tools, in what applications would it be helpful given the definitions of "abductive" and "counterfactual" explanations. The example on the top of page 2 does not convince me in the slightest, given that in modern applications features are basically never equipped with clean semantics as assumed here.
3. The paper starts with some very simple negative results for RNNs and Transformers and then focuses almost entirely on automata models (DFA, NFA, and Interval Automata) that have an unclear connection to most work in ICLR.

Suggestion how to make this work more relevant: The negative results here seem to (at least partially) rely on the fact that weights are rational numbers. However, in modern models the exact value of these weights does not seem to matter very much, as can be seen by the success of quantization techniques. Hence, a much more interesting model of RNNs and Transformers could be one where weights are binary or chosen from a small range (e.g. a 4-bit representation). In terms of explanations: how hard is it to extract small formulas that represent the output, perhaps with PAC guarantees?

**Questions:**

The paper focuses on DFAs and Interval Automata. How are they related to machine learning? Could you give examples that also motivate the class of explanations you consider in this work?

---

> ### Author Response · Authors · 2023-11-22
>
> Q: The paper focuses on DFAs and Interval Automata. How are they related to machine learning and ICLR?
>
> A: We thank you for your comment. In the new version, we have placed a stronger emphasis on the connection to model generation (learning) and explanation generation. Note that our results concern the latter, so we mention relevant existing results on the former to put the results in contexts. We summarize the connection below.
>
> Our work was actually very much motivated by the machine learning work on the extraction of DFA from RNN using automata learning algorithms (e.g. Angluin's L*), e.g., see the ICML paper by Weiss et al. (2018) and the AAAI paper by Okudono et al. (2020). This work hinges on the assumption that DFA is "more interpretable" than RNN, but it was never really clear what this means. Following the work for non-sequential models, we associate interpretability with the computational tractability (following standard complexity theoretic assumptions) to compute abductive and counterfactual explanations. In this technical sense, we showed that DFA is more interpretable than RNN and transformers.
>
> So, a fuller working (F)XAI schema for could be as follows: (1) train RNN R, (2) extract DFA A out of R (e.g. using Weiss et al.), and (3) use our algorithm on A to generate explanations.
>
> Note that (1) could perhaps be replaced directly by an algorithm that extracts a DFA A directly from the datasets. Note also that although we deal with decision problems, the polynomial-time algorithms can be used to extract an explanation (we have clarified this in the new version).
>
> The aforementioned work so far concentrates over strings (over a finite alphabet). Since time series (e.g. stock price chart) is ubiquitous in multiple applications, we went one step further to look at sequential models that may take "data sequences" as input (i.e. sequences of rationals). Two prominent extensions of DFA in this setting are interval automata and k-deterministic register automata, for both of which automata learning algorithms (e.g. Moerman et al. (2017); Howar et al. (2012); Cassel et al. (2016); Drews & D’Antoni (2017); Argyros & D’Antoni (2018)) have been developed in the last 10 years. In the new version, we have also expanded our small examples in the previous version on stock price signals, and how they can be captured by Interval Automata and 2-DRA. Finally, our results are a fundamental building block that enables a similar working schema to the aforementioned case of finite alphabet.
>
> Q: Could you give examples that also motivate the class of explanations you consider in this work?
>
> A: In the revised version, we have also expanded our small examples in the previous version (Example 1 to Example 3). They concern a simple document identification task (using DFA over finite alphabets) and a simple stock trading signal (using interval automata/k-DRA). We show how counterfactual/abductive explanations mean in this context.
>
> Q: Features are never equipped with clearly defined semantics
> A: We agree with this. That said, we would like to convince you that regular languages (and interval automata, as well as k-DRA, in the case of data sequences) are still reasonable representations of features. Firstly, they are well-known to be sufficiently expressive to capture a variety of concepts in many applications. Secondly, a regular language representing a specific feature can itself be learned using an automata learning algorithm. For example, we have a DFA A that classifies whether an input is a spam. Features could be: (1) a DFA B that identifies if the email sender asks for money, and (2) a DFA C that identifies if there are dangerous links (a huge list of known dangerous links). Even when B does not have a clear semantics, it is possible to infer B from examples using a learning algorithm.
>
> Q: What about weights with a few bits? Extracting representations of the output with PAC guarantees?
>
> A: Thank you for this interesting idea for future work. In terms of explainability, the results from Barcelo et al. (2020) showed that small weights (with a fixed number of bits) are sufficient for obtaining NP-hardness for multilayer perceptrons (especially,  MinimumChangeRequired). This NP lower bound carries over to RNN and Transformers. As you correctly pointed out, the proofs of other negative results no longer hold (e.g. Turing-completeness of RNN and Transformers), so a better upper bound could be possible. In terms of a generation of a simplified model (or an explanation), the same working schema that we mentioned above (RNN -> Automata -> explanation) could still be used in this case, but indeed it would be interesting in the future to provide a more solid theoretical guarantee of such a method. [Incidentally, works on extraction of DFA from RNN using ideas of PAC-learning and Angluin's stochastic implementation of equivalence queries are available, e.g., Mayr and Yovine (2018,2021).]

---

### Official Review · Reviewer_56Yr · 2023-11-10

**Soundness:** 3 good
**Presentation:** 3 good
**Contribution:** 2 fair
**Rating:** 6
**Confidence:** 4

**Summary:**

This is a quick review given on a short notice. I did not check in details the
proofs in the appendix.

This paper studies the complexity of two types of interpretability queries, in
the setting of formal interpretable AI (FXAI):
- Computing “minimal sufficient reasons”, also called “abductive explanations”:
  for a given model M and input x over a particular set of features, compute a
  minimal set of features such that setting their values to those of x suffices
  to obtain the same classification as x (i.e., all completions are in the same
  class as x)
- Computing “counterfactual explanations” also called “minimum change required”.
  Here we want to know what is the minimum number of feature values to change
  so that M changes its mind on x. In more generality, given a metric m over
  the possible inputs, one want to find a closest input x to y according to this
  metric so that y and x are not classified the same.

These queries have already been studied, but not for sequential models, which
this paper does. The models that it considers are RNNs, Transformers, and
variants of finite state automata (FSA) for sequences of rational numbers, with
ranges of rationals as transitions (called interval automata), and possibly
with registers that can do comparison (k-DFAs if the automata are
deterministic and k registers, k-NFAs if nondeterministic). The features are
regular languages. Metrics considered are Hamming distance, Edit distance, and
Dynamic Time Warping Distance (DTW).

The results are:

* For RNNs and Transformers
1. minimal sufficient reason is undecidable
2. checking the existence of a counterfactual explanation is undecidable for
   DTW, and NP-complete for Hamming and Edit distance

* For FSAs:
3. Checking the existence of a minimal sufficient reason is PSPACE hard
4. For a fixed k, computing the minimal distance of a counterfactual
   explanation is PTIME over k-DFAs, for all distances mentioned above
5. computing the minimal distance of a counterfactual explanation over k-NFAs
   is NP-hard for Hamming and edit distance, and PSPACE for DWT.

**Strengths:**

The idea of using complexity theory to formally compare the interpretability of
various queries over various models is not new, but the originality here is in
studying FXAI interpretability queries for sequential models instead of models
that only accept inputs of a given length. The idea of using regular languages
for the features is interesting, but it is not clear if this is something new
that the authors are the first to propose, or if this has been already
considered elsewhere. The paper is overall well written and relatively easy to
read (modulo the typos). The topic and results are definitely of interest for
ICLR. I found the proof technique for result 4 is interesting.

**Weaknesses:**

*********** Propositon 3 seems to directly contradict Proposition 2: interval automata
  are more general than DFA, generating is harder than deciding the
  existence... In fact the proof of Proposition 3 stops in the middle, and I could not
  locate the proof in the appendix  ****************
This has to be fixed. This is why I put a soundness score of 2, but if it was only a mistake
from the authors with their latex files during submission and is corrected the soundness score could go higher
--------> This has been fixed in the updated version

- results 1, 2 and 3 listed above seem to me to be straightforward translations of
  existing folklore results. While I agree that there is value in translating and stating these results
  into the world of FXAI, I find it weird to state these as contribution, theorem, etc, when
  they could simply be observations that derive from existing work

- p2 "the same problems were shown to be solvable in polynomial-time for
  decision trees and perceptrons Barceló et al. (2020)": This is misleading.
  According to Barceló et al. (2020), proposition 6, minimum sufficient reasons
  is NP-hard for decision trees.

**Questions:**

- Beware that minimUM and minimAL sufficient reasons might no have the same
  complexity: for instance minimUM is NP-hard over decision trees, but minimAL
  is trivially in linear time for decision trees. In the introduction you mention minimUM, but in
  the rest of the paper you seem to use minimAL everywhere. This should be clarified (see also the comment
on NP-hardness of minimum vs ptime of minimal for decision trees)

- p4 why do you forbid an abductive explanation to be empty? I think that your
  definition is broken and should be fixed as follows: it is simply a minimal
  (under inclusion) subset of features such that v satisfies all features and all
  completions are classified the same. If this set turns out to be empty, this
  just means that the model is constant, and so the "empty" explanation simply
  explains this.

- Proposition 2, and in general: you could also consider the setting where the
  set of features is fixed, which would give you PTIME for DFAs.

I do not really see the point in spending the
  space to define RNNs in the main text since you are not really using it in the
  end... This could be hidden in the appendix

Typos and minors
- In the abstract, first "DFA", then "deterministic finite automata (DFA)": swap
- p5 "that there exists an RNN over...”: existence is not enough I think, you
  need to be able to compute it.
- p6 "let \Psi be any intervals": you mean the set of intervals?
- p6 definition of a run: P is not and should be quantified
- p6 "An interval automaton": missing verb
- p6 "is a sequence of numbers in the range..." -> "is the set of sequences of numbers in the range..."
- p6 "of interests" -> interest
- p6 "The  complexity of a counterfactual query": what does this mean? checking
  existence? checking if there is one with distance \leq k with k given as
  input?
- p7 "(\infty, \infty)" -> ( - \infty, \infty)
- p9 "RNN and Transformers are either difficult": English

---

> ### Author Response · Authors · 2023-11-14
> **Asking for clarification of what you mean by "Results 1, 2, and 3"**
>
> Many thanks for the helpful feedback. Could you please quickly clarify what you mean by Results #1, #2, and #3 "listed above"? We did not see any list with this numbering in your review, and in our paper Propositions and Theorems have different counters. We would not be able to accurately respond without your clarification. Many thanks in advance.

---

> > ### Comment · Reviewer_56Yr · 2023-11-15
> > **Added missing contributions summary**
> >
> > Sorry, I misscopy-pasted my text for the "summary" box and some part was missing, I have updated the review

---

> ### Author Response · Authors · 2023-11-22
>
> Q: Propositon 3 seems to directly contradict Proposition 2 … interval automata are more general than DFA, generating is harder than deciding the existence... In fact the proof of Proposition 3 stops in the middle, and I could not locate the proof in the appendix **************** This has to be fixed. This is why I put a soundness score of 2, but if it was only a mistake from the authors with their latex files during submission and is corrected the soundness score could go higher
>
> A: Thanks for noticing this. There appeared to be a problem with a LaTeX macro that caused this part (Proposition 3 and incompleted proof) to be displayed when we switched into a submission mode. Certainly, this was not meant to be there, which we have deleted in the revised version. We sincerely apologize for the confusion.
>
> Q: Is the use of regular languages for features original?
>
> A: We did not mean to claim this to be our invention, but rather we believe that this is a natural choice from a theoretical perspective (owing to the central role of regular languages). In addition, from a practical perspective, regular languages subsume some standard notions of features for textual data (e.g. the existence of certain substrings in a text like "Republicans" and "Democrats"). We have rephrased this in the revised version.
>
> Q: Results 1, 2 and 3 listed above seem to me to be straightforward translations of existing folklore results. While I agree that there is value in translating and stating these results into the world of FXAI, I find it weird to state these as contribution, theorem, etc, when they could simply be observations that derive from existing work
>
> A: We indicated in the paper that these are not the main contributions, e.g., in the Intro, "We begin by observing …" (instead of "proving"). In addition, results 1, 2 and 3 are stated as "Propositions", instead of "Theorems" (our main results).
>
> Our contribution for results 1, 2, and 3 is to *observe* that they follow from the existing results, sometimes with a bit of work (the proof sometimes requires a long and tedious encoding, e.g., see Appendix). To the best of our knowledge, these results were never stated, mentioned, and proved anywhere, which is why we state them and provide a proof of them in the appendix for the sake of completeness. In the revised version, we have tried to further emphasize that these are not our main contributions by mentioning "straightforward applications" in the introduction.
>
> Q: Beware that minimUM and minimAL sufficient reasons might no have the same complexity: for instance minimUM is NP-hard over decision trees, but minimAL is trivially in linear time for decision trees. In the introduction you mention minimUM, but in the rest of the paper you seem to use minimAL everywhere. This should be clarified (see also the comment on NP-hardness of minimum vs ptime of minimal for decision trees)
>
> A: Thank you for noticing this. You were right that we meant *minimal* sufficient reasons throughout the paper, which is ptime solvable for decision trees (but is NP-hard for decision lists). We have corrected this in the revised version.
>
>
> Q: p2 "the same problems were shown to be solvable in polynomial-time for decision trees and perceptrons Barceló et al. (2020)": This is misleading. According to Barceló et al. (2020), proposition 6, minimum sufficient reasons is NP-hard for decision trees.
>
> A: We thank you for noticing this and apologize for the confusion. Indeed, minimum sufficient reasons are NP-hard for decision trees, whereas minimal sufficient reasons are polynomial-time solvable (which is the problem we deal with in the paper). We have corrected the citations in the revised version.
>
>
>
> Q: p4 why do you forbid an abductive explanation to be empty? I think that your definition is broken and should be fixed as follows: it is simply a minimal (under inclusion) subset of features such that v satisfies all features and all completions are classified the same. If this set turns out to be empty, this just means that the model is constant, and so the "empty" explanation simply explains this.
> A: We have incorporated your more general definition in the revised version, noting that an empty intersection results in \Sigma^*. The PSPACE-hardness result holds either way.
>
> Q: Proposition 2, and in general: you could also consider the setting where the set of features is fixed, which would give you PTIME for DFAs.
>
> A: Thanks for the suggestion. We have added this statement in the text.
>
> Q: I do not really see the point in spending the space to define RNNs in the main text since you are not really using it in the end... This could be hidden in the appendix
>
> A: We have relegated this to the appendix in the new version.

---

> ### Comment · Reviewer_56Yr · 2023-11-23
> **Answer to authors' answer**
>
> Thank you, I am overall a satisfied reviewer with the answers your made to my comments, and with the new version. Now that the problem with Proposition 3 is fixed I will update the soundness score from 2 to 3 (not 4 because I did not have time to check all the proofs).
>
> That being said, I will not change the final score of my review. Indeed, I still consider that the only place where something technically nontrivial happens in the paper is Theorem 2 from the current version, and that the other results are rather decorative and are primarily here to form a nice and coherent story. It *is* a nice story though, and the paper is nicely written and easy to follow, and by contrast to other reviewers I do not care at all that there are no experiments (but I am not very familiar with ICLR so I will leave this to the appreciation of the PC), so that is why I put "marginally above acceptance threshold". But I am not ready to go higher for that reason.
>
> Last comment:
> In the updated draft, the definition of abductive explanations still includes "and that M (w) \neq M (v) for some sequence
> w satisfying all features in any strict subset S' subseteq S", which should be removed.

---

> > ### Author Response · Authors · 2023-11-23
> >
> > Many thanks again for the numerous helpful and constructive feedback, which certainly has improved our paper.
> >
> > Yes, in terms of difficulty of proofs, the proof of Theorem 2 is by far the most challenging. We think that introducing this automata model in the context of explainability for sequential models in ML for data sequences (e.g. time series) is also one of the main contributions of the paper, since register automata have never made its appearance in the ML community (in fact, the only application thereof we have seen so far is the graph database community - see JACM'16 paper by Libkin, Martens, and Vrgoc), but they can naturally be used as representations for time series data.
> >
> > Thanks for the "last comment" - we have implemented this correction in the new revised version.

---

### Author Response · Authors · 2023-11-22
**Summary of rebuttal and changes**

We thank you for your work and the helpful comments. We have responded to individual reviewer's questions by official comment to the specific reviewer. Many of these helpful comments and questions have been incorporated into the revised version. We summarize the changes here:
1. We have expanded the examples explicating the uses of DFA, interval automata, and k-DRA that we gave in the previous version. These examples are from document classification and stock trading strategies. See Example 1, Example 2, and Example 3. Here, we also discuss what abductive and counterfactual explanations mean in these contexts.
2. To fit everything into 9 pages, we have relegated proof sketch of Proposition 1 and proof of Proposition 2 into the appendix.
3. We have placed more emphasis on model generation and explanation generation in the text. This is done by emphasizing the connections to existing automata learning algorithms, model extraction from more complex models (e.g. RNN), and last but not least our polynomial-time algorithms in fact easily enable the generation of an explanation.
4. There appeared to be a problem with a LaTeX macro that caused Proposition 3 and incompleted proof to be displayed when we switched into a submission mode. We have deleted in the revised version. We sincerely apologize for the confusion, and thank you for pinpointing this problem.

---

### Meta-Review · Area_Chair_b2o3 · 2023-12-05

**Metareview:**

The paper is about the (post-hoc) explainability of sequential models with tools from automata theory.
The main idea is to follow the approach (given for instance in Weiss et al 2018) to build a DFA from RNN using automata learning algorithms. The DFA is supposed to be more explainable than RNNs. The contribution is to provide abductive and conterfactual explanations using polynomial-time algorithms for deterministic interval automata and deterministic register automata with a fixed number of registers.

The contribution is purely theoretical and no experimental evidence is provided about the "utility" of the computed explanations. The sequence of transformations used to generate these explanations reinforces doubts about their relevance.

From the theoretical point of view, the key contribution is in Thm 2. The result is non-trivial. It is firmly rooted in language theory and may not find its audience in ICLR. The paper is certainly better suited to other venues.

**Justification For Why Not Higher Score:**

There is some decent work in the paper, but the main results concern language theory and there is no evidence that the method can provide useful explanations.

**Justification For Why Not Lower Score:**

N/A

---

### Decision · Program_Chairs · 2024-01-16

Reject